# Aberration correction in long GRIN lens-based microendoscopes for extended field-of-view two-photon imaging in deep brain regions

Andrea Sattin[1,2†], Chiara Nardin[1,2†], Simon Daste[3], Monica Moroni[2,4], Innem Reddy[5], Carlo Liberale[5], Stefano Panzeri[2,6], Alexander Fleischmann[3], Tommaso Fellin[1,2]*

[1]Optical Approaches to Brain Function Laboratory, Istituto Italiano di Tecnologia, Genova, Italy; [2]Neural Coding Laboratory, Istituto Italiano di Tecnologia, Genova and Rovereto, Italy; [3]Department of Neuroscience and Carney Institute for Brain Science, Brown University, Providence, United States; [4]Neural Computation Laboratory, Center for Neuroscience and Cognitive Systems @UniTn, Istituto Italiano di Tecnologia, Rovereto, Italy; [5]Biological and Environmental Sciences and Engineering Division (BESE), King Abdullah University of Science and Technology (KAUST), Thuwal, Saudi Arabia; [6]Institute for Neural Information Processing, Center for Molecular Neurobiology (ZMNH), University Medical Center Hamburg-Eppendorf (UKE), Hamburg, Germany

*For correspondence: tommaso.fellin@iit.it

†These authors contributed equally to this work

Competing interest: The authors declare that no competing interests exist.

## eLife Assessment

This **valuable** study builds on previous work by the authors by presenting a potentially key method for correcting optical aberrations in GRIN lens-based microendoscopes used for imaging deep brain regions. By combining simulations and experiments, the authors provide **convincing** evidence showing that the obtained field of view is significantly increased with corrected, versus uncorrected microendoscopes. Because the approach described in this paper does not require any microscope or software modifications, it can be readily adopted by neuroscientists who wish to image neuronal activity deep in the brain.

**Abstract** Two-photon (2P) fluorescence imaging through gradient index (GRIN) lens-based endoscopes is fundamental to investigate the functional properties of neural populations in deep brain circuits. However, GRIN lenses have intrinsic optical aberrations, which severely degrade their imaging performance. GRIN aberrations decrease the signal-to-noise ratio (SNR) and spatial resolution of fluorescence signals, especially in lateral portions of the field-of-view (FOV), leading to restricted FOV and smaller number of recorded neurons. This is especially relevant for GRIN lenses of several millimeters in length, which are needed to reach the deeper regions of the rodent brain. We have previously demonstrated a novel method to enlarge the FOV and improve the spatial resolution of 2P microendoscopes based on GRIN lenses of length <4.1 mm (Antonini et al., 2020). However, previously developed microendoscopes were too short to reach the most ventral regions of the mouse brain. In this study, we combined optical simulations with fabrication of aspherical polymer microlenses through three-dimensional (3D) microprinting to correct for optical aberrations in long (length >6 mm) GRIN lens-based microendoscopes (diameter, 500 μm). Long corrected microendoscopes had improved spatial resolution, enabling imaging in significantly enlarged FOVs.

Moreover, using synthetic calcium data we showed that aberration correction enabled detection of cells with higher SNR of fluorescent signals and decreased cross-contamination between neurons. Finally, we applied long corrected microendoscopes to perform large-scale and high-precision recordings of calcium signals in populations of neurons in the olfactory cortex, a brain region laying approximately 5 mm from the brain surface, of awake head-fixed mice. Long corrected microendoscopes are powerful new tools enabling population imaging with unprecedented large FOV and high spatial resolution in the most ventral regions of the mouse brain.

## Introduction

High-resolution 2P fluorescence imaging of the awake brain is a fundamental tool to investigate the relationship between the structure and the function of brain circuits (*Svoboda and Yasuda, 2006*). Compared to electrophysiological techniques, functional imaging in combination with genetically encoded indicators allows monitoring the activity of genetically targeted cell types, access to subcellular compartments, and tracking the dynamics of many biochemical signals in the brain (*Xu et al., 2024*). However, a critical limitation of multiphoton microscopy lies in its limited (<1 mm) penetration depth in scattering biological media (*Helmchen and Denk, 2005*).

Deep ventral regions of the brain, such as the olfactory cortex, the hypothalamus, and the amygdala play fundamental roles in controlling the processing of sensory information (*Endo and Kazama, 2022*), circadian rhythms (*Sternson, 2013*), hormone release (*Sternson, 2013*), decision making (*Phelps et al., 2014*), and adaptation of instinctive and motivational behavior (*Phelps et al., 2014*). Since these ventral areas lay deeper than 4 mm down the brain surface in rodents and larger mammals (e.g. rats, cats, and marmosets), imaging these regions typically requires the use of light guides, which relay the focal plane of the microscope objective down to the target brain area. Since inserting light guides in the brain damages the tissue above the target area (*Antonini et al., 2020*), reducing the cross-section of the probe is desired when imaging these deep brain regions. One popular solution for deep 2P imaging in awake animals is using GRIN lenses (*Jung and Schnitzer, 2003*; *Jung et al., 2004*; *Reed et al., 2002*; *Barretto et al., 2009*), thin cylindrical rod lenses characterized by refractive index which varies with the radial distance (*Moore, 1980*). For example, in the mouse orbitofrontal cortex 2P microscopy through GRIN endoscopes showed that orbitofrontal neurons respond to either caloric rewards or social stimuli, indicating the presence of functionally distinct feeding and social neuronal subnetworks in this deep cortical region (*Jennings et al., 2019*). Similarly, in the lateral hypothalamus combining GRIN endoscopes with 2P imaging revealed neurons encoding thermal punishment and reward and this subset of neurons was distinct from the ensemble of neurons encoding caloric reward (*Jung et al., 2022*). Other ventral brain regions, such as the olfactory cortex, can be imaged with an invasive preparation, which does not require a GRIN lens, but which is incompatible with imaging in awake behaving animals (*Roland et al., 2017*; *Pashkovski et al., 2020*). In the olfactory cortex, GRIN lens-based 2P endoscopy becomes necessary in awake animals (*Wang et al., 2020*), where head tilting is not compatible with animal performance in a behavioral task.

One limitation of GRIN lens-based endoscopy is that GRIN lenses suffer from intrinsic optical aberrations (*Bortoletto et al., 2011*; *Lee and Yun, 2011*). GRIN aberrations can be both on-axis and off-axis and severely degrade the quality of 2P imaging. GRIN aberrations distort the point spread function (PSF) and enlarge the excitation volume of the focal spot, leading to reduced SNR of fluorescence signals and decreased spatial resolution (*Antonini et al., 2020*; *Wang and Ji, 2012*; *Wang and Ji, 2013*). This effect is not uniform across the FOV and is particularly relevant in lateral portions of the FOV, generating uneven spatial resolution and inhomogeneous amplitude of fluorescence signals across the FOV (*Antonini et al., 2020*; *Wang and Ji, 2012*; *Wang and Ji, 2013*). As a consequence of these limitations, previous studies using 2P imaging through GRIN lenses in ventral regions of the mouse brain were limited to imaging a small number of neurons with uneven spatial resolution across the FOV and low SNR in lateral portion of the FOV (*Jennings et al., 2019*; *Jung et al., 2022*; *Piantadosi et al., 2024*). Developing GRIN-based endoscopic probes with improved optical properties and more homogeneous spatial resolution, which are long enough to enable imaging in the most ventral regions of the rodent brain while maintaining a small (≤500 μm) cross-section, is thus a compelling need.

Various adaptive optics methods were developed to correct optical aberrations in GRIN lenses. For example, initial efforts utilized optical compensation by low-order electrostatic membrane mirror (*Bortoletto et al., 2011*). Alternatively, adaptive optics through pupil segmentation was used to efficiently correct GRIN aberrations, increasing the intensity of recorded fluorescence signals and enlarging the imaging FOV (*Wang and Ji, 2012*; *Wang and Ji, 2013*). More recently, geometric transformation adaptive optics has been developed and applied for the correction of aberrations in GRIN lenses (*Li et al., 2024*). However, adaptive optics approaches require substantial modification of the optical path of the microscope (*Bortoletto et al., 2011*; *Wang and Ji, 2012*; *Wang and Ji, 2013*; *Li et al., 2024*), needs specifically designed software controls (*Wang and Ji, 2012*; *Wang and Ji, 2013*; *Li et al., 2024*), and may limit the temporal resolution of imaging (*Wang and Ji, 2012*; *Wang and Ji, 2013*). Alternatively, mm-size plano-convex lenses have been combined with GRIN lenses to correct on-axis aberrations and increase the Numerical Aperture (NA) of the optical system (*Barretto et al., 2009*). However, this method is difficult to apply to GRIN lenses of cross-section smaller than 1 mm (*Matz et al., 2015*), because of the difficulty in manufacturing high-precision optics with small radial dimension and complex profile. Recently, a novel method based on 3D microprinting of polymer optics was developed to correct for GRIN aberrations by placing specifically designed aspherical corrective lenses at the back end of the GRIN lens (*Antonini et al., 2020*). This approach is attractive because it is built-in on the GRIN lens and corrected microendoscopes are ready-to-use, requiring no change in the optical set-up. However, previous work demonstrated the feasibility of this method only for GRIN lenses of length <4.1 mm (*Antonini et al., 2020*), which are too short to reach the most ventral regions of the mouse brain. The applicability of this technology to longer GRIN lenses, which are affected by stronger optical aberrations (*Lee and Yun, 2011*), remained to be proven.

In this study, we designed, developed, and validated correction of optical aberrations in GRIN lenses of length 6.4 mm and 8.8 mm and diameter 500 μm using a combination of ray-trace simulation, 3D microprinting by 2P lithography, simulated calcium data, and experimental characterization. Validation of the new corrected microendoscopes is provided with proof-of-principle experiments in a ventral brain region, the olfactory cortex, of awake head-fixed mice.

## Results
### Design of aberration corrected microendoscopic probes based on long GRIN rods

We selected two GRIN singlet rod lenses (hereon GRIN rods) suitable for multiphoton fluorescence imaging in ventral regions of the mouse brain. The two selected GRIN rods had NA = 0.5 on both sides, 500 μm diameter, 1.5 and 2.0 pitch, and 6.4 mm and 8.8 mm length, respectively. For both GRIN rods, we modeled an optical assembly composed of the GRIN rod, a thin layer of glass (representing a 100-μm-thick glass coverslip) attached to the back end of the GRIN rod, and an aspheric corrective lens (here on called corrective lens) aligned to the rod and attached to the opposite side of the glass coverslip with respect to the GRIN rod (*Figure 1A and B*). We designed the aspheric surface of the corrective lens using ray-trace simulations at the wavelength ($\lambda_{exc}$) of 920 nm, the excitation wavelength commonly used for 2P imaging of the popular genetically encoded calcium indicator GCaMP. The design of the corrective lens was an iterative process in which the parameters that define the corrective lens surface were automatically varied to obtain the highest and most homogeneous spatial resolution possible over the longest possible radial distance from the optical axis (see *Supplementary file 1* and Materials and methods for details on simulation parameters). More specifically, we simulated the corrective lens profile such that the Strehl ratio (defined as the peak intensity of the focal spot of an aberrated optical system normalized to the peak intensity of the focal spot of the ideal optical system, *Smith, 2008*) of the corrected microendoscope remained above the threshold of 0.8 for the longest possible radial distance from the optical axis. The values of Strehl ratio equal to 0.8 corresponded to the lower bound of the diffraction-limited condition set by the Maréchal criterion (*Antonini et al., 2020*; *Smith, 2008*). Our simulations showed that the profiles of the two corrective lenses displayed in *Figure 1A and B* led to an increase in the radius of the diffraction-limited FOV compared to the case in which the corrective lens was not positioned in the back end of the GRIN rods (uncorrected microendoscopes). The profile of the corrective optical element was specific to the type of GRIN rod considered. The simulated increase in the radius of the diffraction-limited FOV was 3.50

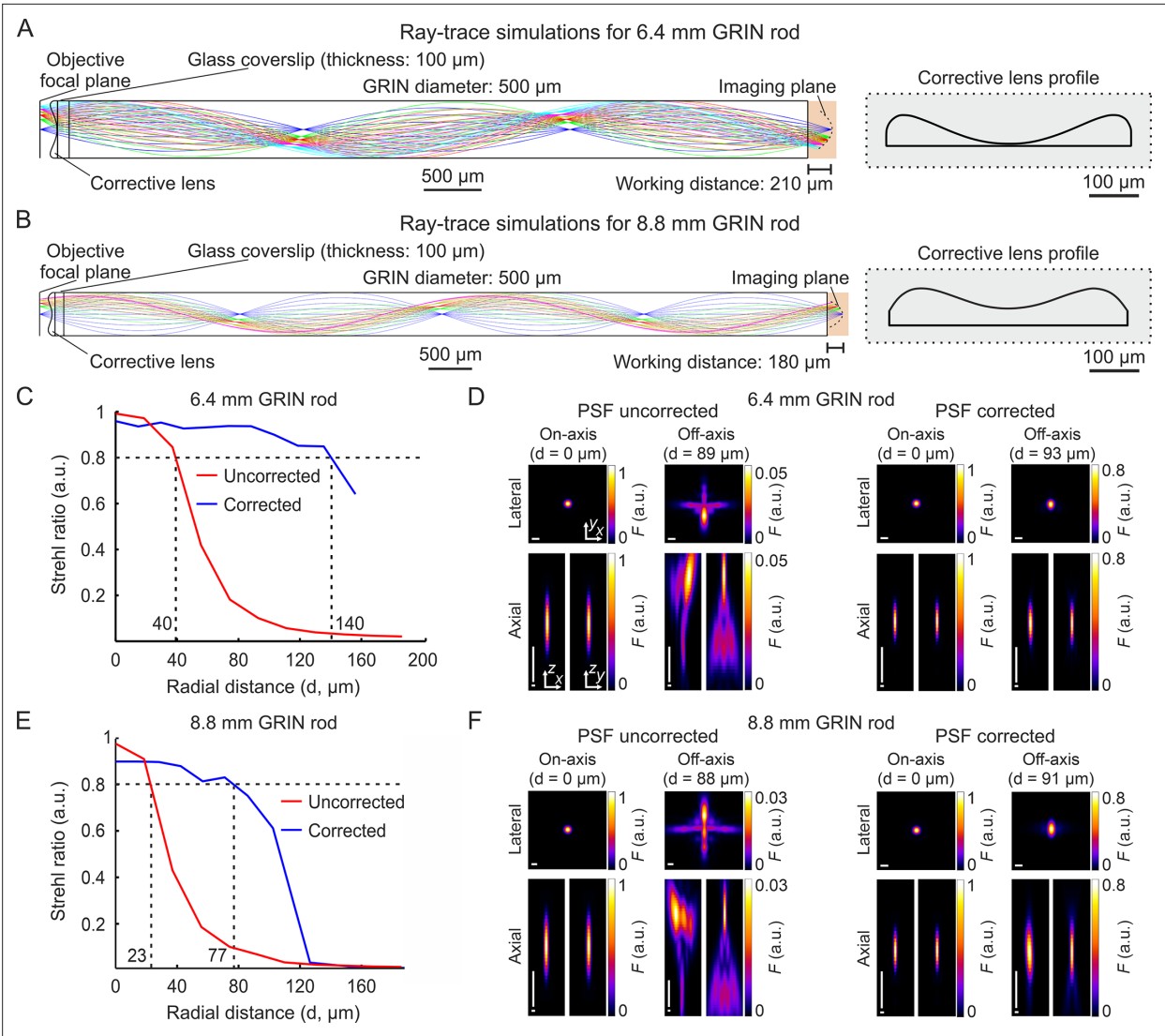

**Figure 1.** Optical simulations of long corrected microendoscopes. (**A**) Ray-trace simulation for the microendoscope based on the 6.4 mm-long GRIN rod. Left: rays of 920 nm light are relayed from the objective focal plane to the imaging plane. Labels indicate geometrical parameters of the microendoscope components. Right: profile of the corrective aspherical lens that maximizes the microendoscope FOV in the optical simulation shown on the left. (**B**) Same as (**A**) for the microendoscope based on the 8.8 mm-long GRIN rod. (**C,D**) Optical performance of simulated microendoscope based on the 6.4 mm-long GRIN rod. (**C**) Strehl ratio as a function of the field radial distance (zero indicates the optical axis) computed on the focal plane in the object space (after the GRIN rod) for the corrected (blue) and the uncorrected (red) microendoscope. The black horizontal dashed line indicates the diffraction-limited threshold according to the Maréchal criterion (**Smith, 2008**). The black vertical dashed lines mark the abscissa values of the intersections between the curves and the diffraction-limited threshold. (**D**) Lateral ($x,y$) and axial ($x,z$ and $y,z$) intensity profiles of simulated PSFs on-axis (distance from the center of the FOV d=0 µm) and off-axis (at the indicated distance d) for the uncorrected (left) and the corrected microendoscope (right). Horizontal scale bars: 1 µm; vertical scale bars: 10 µm. (**E,F**) Same as (**C,D**) for the microendoscope based on the 8.8 mm-long GRIN rod.

The online version of this article includes the following source data and figure supplement(s) for figure 1:

**Source data 1.** Numerical values to reproduce graphs in *Figure 1C and E*.

**Figure supplement 1.** Performance of corrected microendoscopes at different wavelengths.

**Figure supplement 2.** Optical performance in out-of-focus planes.

**Figure supplement 3.** Optical performance of corrected microendoscopes as a function of decentering the corrective lens.

times and 3.35 times for the 6.4 mm-long and 8.8 mm-long probe, respectively (*Figure 1C and E*). Moreover, we investigated the effect of changing wavelength on the Strehl ratio. We found that the Strehl ratio remained >0.8 within at least ±10 nm from $\lambda_{exc}$ = 920 nm (*Figure 1—figure supplement 1*), which covered the limited bandwidth of our femtosecond laser. All simulations were performed

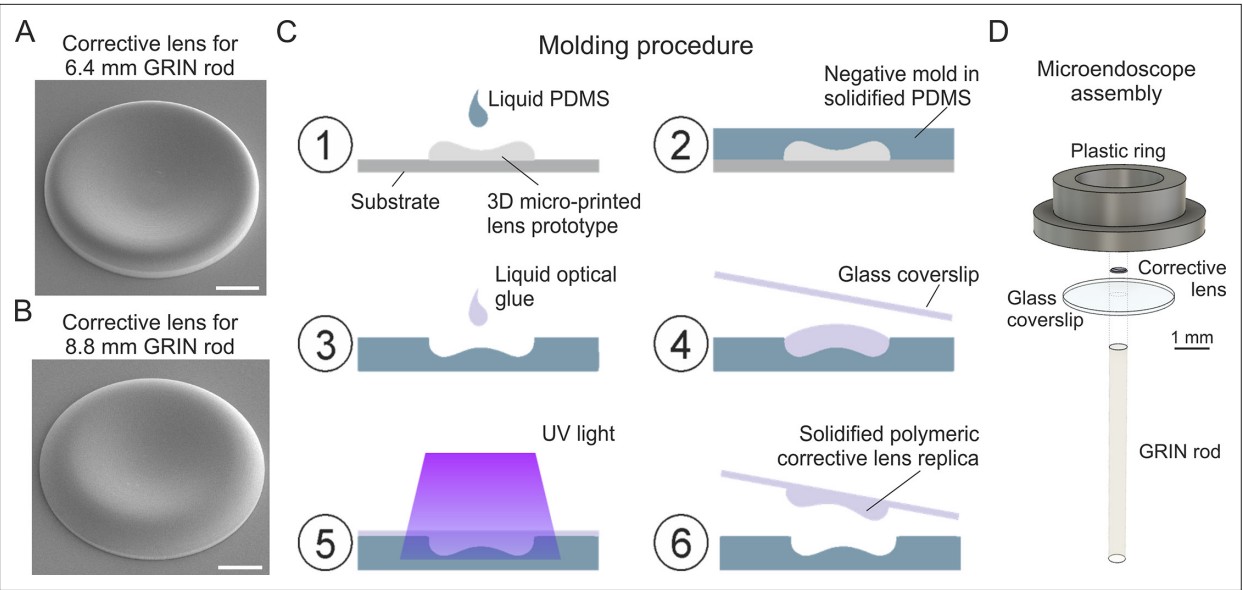

**Figure 2.** Fabrication of corrective lenses using 3D microprinting and assembly of long microendoscopes. (**A**) Scanning electron microscopy image of the 3D microprinted replica of the corrective lens for the 6.4 mm-long corrected microendoscope. Scale bar: 100 µm. (**B**) Same as (**A**) for the 8.8 mm-long microendoscope. (**C**) Molding procedure for the generation of corrective lens replica. Freshly prepared PDMS is casted onto a corrective aspherical lens printed with 2P lithography (1). After 48 hr, the solidified PDMS provides a negative mold for the generation of lens replica (2). A small drop of optical UV-curable glue is deposited onto the mold (3). The mold filled by optical glue is covered with a glass coverslip, which is gently pressed against the mold (4). The optical glue is polymerized with UV light (5). The coverslip with the attached polymeric lens replica is detached from the negative mold (6). Object dimensions are not to scale. (**D**) Exploded view of the corrected microendoscope assembly.

maximizing aberration correction in the simulated focal plane of the microendoscopes (imaging plane in *Figure 1A and B*). However, we explored the effect of aberration correction outside the simulated focal plane. In corrected microendoscopes, we found that for off-axis rays (radial distance from the optical axis >40 µm) the Strehl ratio was >0.8 (Maréchal criterion) in a larger volume compared to uncorrected microendoscopes (*Figure 1—figure supplement 2*), demonstrating that the aberration correction method developed in this study extends for short distances beyond the simulated focal plane. For example, at a radial distance of ~90 µm from the optical axis, the axial range in which the Strehl ratio was >0.8 in corrected microendoscopes was 28 µm and 19 µm for the 6.4 mm-long and the 8.8 mm-long microendoscope, respectively. Decentering the optical axis of the corrective lens from the optical axis of the GRIN rod rapidly disrupted the optical properties of the corrected microendoscopes (*Figure 1—figure supplement 3*). To visualize the expected improvement of the spatial resolution in corrected microendoscopes, we simulated the PSF of uncorrected and corrected microendoscopes at different radial distances from the optical axis. For both the 6.4 mm-long and 8.8 mm-long GRIN rods, we found that the PSF of the uncorrected probe was strongly aberrated and highly irregular at ~90 µm away from the center of the FOV (*Figure 1D and F*). In contrast, in the corrected microendoscopes the PSF had regular shape and, at >90 µm away from the optical axis, it was similar to the one measured at the center of the FOV (*Figure 1D and F*, see also *Supplementary file 2*).

## Microfabrication of corrective lenses and assembly of long corrected microendoscopes

Due to the small size of the simulated corrective lenses (diameter: 500 µm; height: few tens of µm), we first used 3D microprinting based on 2P lithography to fabricate aspherical corrective lens prototypes (*Figure 2A and B*; *Liberale et al., 2010*; *Gonzalez-Hernandez et al., 2023*). We then replicated the prototypes using a molding procedure (*Schaap and Bellouard, 2013*; *Figure 2C*). The molding strategy enabled us to obtain large numbers of aspherical lens replica directly manufactured onto round glass coverslips in a much faster and cheaper way compared to 3D microprinting. Next, we assembled microendoscopes following the scheme in *Figure 2D*, using a plain coverslip for the

uncorrected case, while we used a coverslip with a corrective lens replica for the corrected case (this applies to all the experiments described in the manuscript). In the assembly of the microendoscope, we attached the round coverslip to the annular support structure shown in (*Figure 2D*, called hereon ring), which had two functions: (*i*) to facilitate holding the probe during stereotactic surgery implantation; (*ii*) to provide stability to the microendoscope once implanted on the animal skull. We used either metallic rings (external diameter: 7 mm) or plastic rings (external diameter: 4.5 mm). Plastic rings were specifically designed and 3D printed to be compatible with commercial holders (ProView Implant Kit, Inscopix Inc Mountain View, CA, US) to ease stereotaxic implantation (see Materials and methods). The alignment and assembly of all the microendoscope components was performed with a custom optomechanical set-up similar to that described in *Antonini et al., 2020*.

## Improved optical performance of long corrected microendoscopes

We tested the optical performance of the new corrected microendoscopes in 2P laser scanning microscopy (2PLSM) using a pulsed laser tuned at 920 nm. Microendoscopes were coupled to the 2P microscope using a previously described customized mount (*Antonini et al., 2020*). To visualize the effect of the corrective lens on the axial extension of the excitation volume, we imaged subresolved homogenous fluorescent layers (layer thickness: ~300 nm) using uncorrected and corrected microendoscopes. *Figure 3A–D* shows the *x,z* intensity projections of z-stacks acquired through the subresolved fluorescent layers under both conditions (i.e. uncorrected and corrected microendoscopes). Strong off-axis aberrations of the GRIN lens generated a significant loss of fluorescence intensity in the marginal portions of the FOV in uncorrected microendoscopes (for both the 6.4 mm-long, *Figure 3A and B*, and the 8.8 mm-long, *Figure 3C and D*). In contrast, fluorescence signal was collected over longer distances from the center of the FOV in corrected microendoscopes (*Figure 3A–D*). Moreover, for GRIN rods of both lengths the full width at half maximum (FWHM) of the film profile obtained with uncorrected microendoscopes increased with distance from the optical axis and became rapidly larger than 20 μm. In contrast, the film profile obtained with corrected microendoscopes showed more homogenous FWHM values across an extended FOV (*Figure 3A–D*).

To evaluate the increase in area of the sample that could be imaged with the corrected microendoscopes, we first characterized potential distortion of the FOV introduced by the microendoscopes. Using a calibration ruler, we measured the local magnification factor at different radial distances from the optical axis (*Figure 3E and F*). We defined the local magnification factor as the ratio between the nominal pixel size (i.e. pixel size that applies to undistorted images collected with the microscope objective alone) and the real pixel size of the image collected with the same objective coupled to the microendoscope (see Materials and methods). Pixel sizes and distances displayed in the following panels and graphs are calibrated for FOV distortion and indicate real dimensions in the samples (see also Materials and methods and *Supplementary file 3*). We then measured the spatial resolution of microendoscopes by imaging subresolved fluorescent beads (bead diameter: 100 nm) located at different radial distances from the optical axis (*Figure 3G–J*). We found that the axial dimension of the beads imaged with uncorrected microendoscopes rapidly increased with radial distance, whereas the spatial resolution of corrected microendoscopes remained more homogeneous over longer radial distances and more similar to the one measured on-axis (*Figure 3G–J* and *Supplementary file 4*). We set a threshold of 10 μm on the axial resolution to define the radius of the effective FOV (corresponding to the black triangles in *Figure 3I and J*) in uncorrected and corrected microendoscopes. We observed a relative increase of the effective FOV radius of 2.17 and 1.53 for the 6.4 mm-long and the 8.8 mm-long microendoscope, respectively (*Table 1*). This corresponded to an enlargement of the effective FOV area of 4.7 times and 2.3 times for the 6.4 mm-long microendoscope and the 8.8 mm-long microendoscope, respectively (*Table 1*). These findings were in agreement with the results of the ray-trace simulations (*Figure 1*) and the measurement of the subresolved fluorescence layers (*Figure 3A–D*).

The optical characterization described above indicated that corrected microendoscopes of both lengths enabled imaging a larger FOV area compared to uncorrected microendoscopes. This result can be explained by the fact that corrected microendoscopes showed higher spatial resolution and increased probability of 2P fluorescence excitation in the marginal portions of the FOV. Corrected microendoscopes should therefore allow highly contrasted imaging of cell bodies and thin cellular processes over an extended FOV compared to uncorrected microendoscopes. Using both

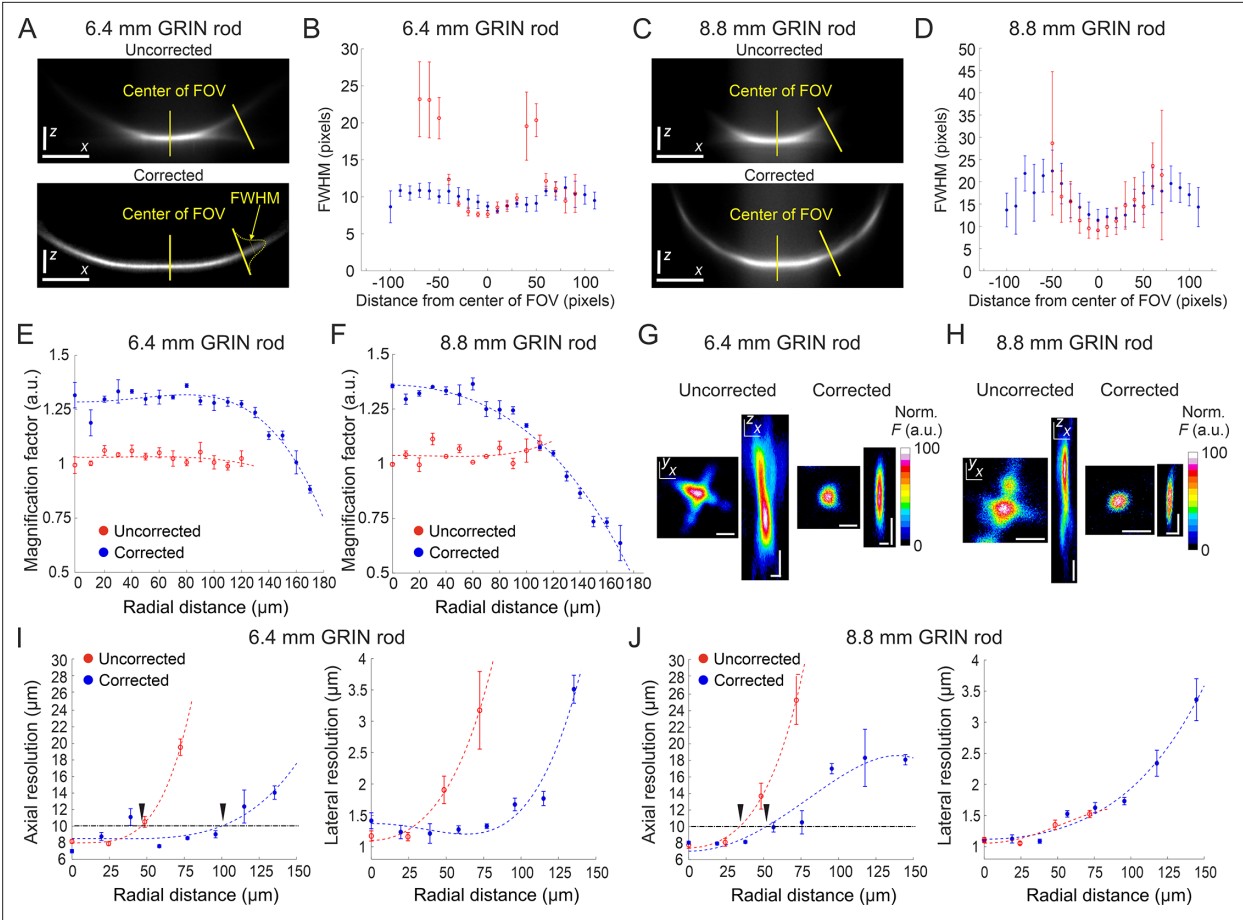

**Figure 3.** Optical characterization of long corrected microendoscopes shows improved spatial resolution over an enlarged FOV. (**A**) Representative *x* (horizontal), *z* (vertical) projection of a z-stack of a subresolved fluorescent layer acquired with the uncorrected (top) or corrected (bottom) microendoscope based on the 6.4 mm-long GRIN rod. $\lambda_{exc}$=920 nm; scale bars: 50 pixels. (**B**) Thickness (mean values ± s.e.m.) of the layer as a function of the distance from the center of the FOV for uncorrected (red, n=4) or corrected (blue, n=4) microendoscopes. The thickness of the film is measured as the FWHM of the Gaussian fit of the fluorescence intensity along segments orthogonal to the tangential line to the section of the film and located at different distances from the center of the FOV (see yellow labels on the bottom image). (**C,D**) Same as (**A,B**) for the microendoscope based on the 8.8 mm-long GRIN rod. (**E**) The distortion of the FOV in uncorrected and corrected microendoscopes is evaluated using a calibration ruler. The magnification factor is defined as the ratio between the nominal and the real pixel size of the image and shown as a function of the radial distance for uncorrected (red, n=3) or corrected (blue, n=3) microendoscopes. Data are shown as mean values ± s.e.m. Fitting curves are quartic functions $f(x)=ax^4+bx^2+c$ (see also *Supplementary file 3* for details). (**F**) Same as (**E**) for the microendoscope based on the 8.8 mm-long GRIN rod. (**G–J**) The spatial resolution of microendoscopes was measured acquiring z-stacks of subresolved fluorescent beads (bead diameter: 100 nm) located at different radial distances using 2PLSM ($\lambda_{exc}$=920 nm). (**G**) Representative *x,y* and *x,z* projections of a fluorescent bead located at a radial distance of 75 μm, imaged through an uncorrected (left) or a corrected (right) 6.4 mm-long microendoscope. Horizontal scale bars, 2 μm; vertical scale bars, 5 μm. (**H**) Same as (**G**) for the microendoscope based on the 8.8 mm-long GRIN rod. (**I**) Axial (left) and lateral (right) resolution (i.e. average size of the *x,z* and *x,y* projections of imaged beads, respectively) as a function of the radial distance from the center of the FOV for uncorrected (red) and corrected (blue) probes. Each data point represents the mean value ± s.e.m. of n=4–24 beads imaged using at least m=3 different 6.4 mm-long microendoscopes. Fitting curves are quartic functions $f(x)=ax4+bx2+c$ (see *Supplementary file 4* for details). The horizontal black dash-dotted line indicates the axial resolution threshold of 10 μm. The black triangles indicate the intersections between the threshold and the curves fitting the data and mark the estimated radius of the effective FOV of the probes. (**J**) Same as (**I**) for the microendoscopes based on the 8.8 mm-long GRIN rod.

The online version of this article includes the following source data for figure 3:

**Source data 1.** Numerical values to reproduce graphs in *Figure 3B, D–F,I and J*.

uncorrected and corrected microendoscopes, we confirmed these predictions by imaging the same region of a fixed brain slice expressing jGCaMP7f in neurons. We found that aberration correction enabled to clearly resolve neuronal somata and processes in the lateral portions of the FOV, whereas in the absence of the corrective lens the same structures located in the peripheral portion of the FOV appeared dim, strongly aberrated, and hardly detectable (*Figure 4*, see also *Figure 4—figure*

**Table 1.** Experimental measurement of enlarged effective FOV in long corrected microendoscopes.
The values of the effective FOV radius for uncorrected and corrected microendoscopes were estimated from the intersection between the arbitrary threshold of 10 μm on the axial resolution and the quartic function fitting the experimental data of *Figure 3I and J*.

| Microendoscope type | Effective FOV radius (μm) | Fold increase in FOV radius | Fold increase in FOV area |
|---|---|---|---|
| 6.4 mm-long GRIN rod | Uncorrected: 46<br>Corrected: 100 | 2.17 | 4.7 |
| 8.8 mm-long GRIN rod | Uncorrected: 34<br>Corrected: 52 | 1.53 | 2.3 |

*supplement 1* for an alternative visualization in which the FOVs of corrected microendoscopes are rescaled to match the real pixel size of the FOVs of uncorrected microendoscopes in the center of the image).

## More accurate sampling of simulated neuronal activity with long corrected microendoscopes

Using simulated calcium t-series, we next evaluated the impact of the improved optical performance of long aberration corrected microendoscopes on the ability to extract information about neuronal activity. To this aim, we compared simulated calcium traces imaged with either uncorrected or corrected microendoscopes with the ground truth neuronal activity, that is the neuronal activity used to generate the simulated t-series. To build synthetic calcium data, we first generated neurons with 3D distribution and anatomical properties (i.e. cell size and cell density) similar to those measured in the mouse olfactory cortex (*Suzuki and Bekkers, 2010*; *Suzuki and Bekkers, 2011*), the ventral brain region which we will be imaging in vivo (see next section of Results), and the biophysical characteristics of one of the two indicators used in our experiments, jGCaMP8f (*Zhang et al., 2023*; see also Materials and methods). We then simulated the optical sampling of 3D volume of neurons with the PSF properties of the uncorrected and corrected microendoscopes characterized above (*Figure 3*). Fluorescence signals integrated over the PSF volume were then projected on a 2D matrix of pixels in time, with a frequency of 30 Hz, to obtain synthetic t-series (duration: 5 min, *Figure 5A and D*). We finally detected neuronal cell bodies and extracted fluorescence traces from regions of interest (ROIs) of synthetic t-series using established methods (*Sità et al., 2022*) commonly applied to real data and compared between the case of uncorrected and corrected microendoscopes. We found that the use of corrected probes allowed segmenting more ROIs with high SNR compared to uncorrected microendoscopes, shifting the distribution of SNR across ROIs to higher mean SNR values. This was true for both the microendoscopes based on the 6.4 mm-long GRIN rod (*Figure 5B*) and the microendoscope based on the 8.8 mm-long GRIN rod (*Figure 5E*). Moreover, for both microendoscope types, aberration correction increased the maximal radial distance at which ROIs with high (>15) SNR could be observed (*Figure 5C and F*). These findings demonstrate that correcting for optical aberrations increases the probability of recording ROIs with larger SNR of fluorescent signals.

The higher and more homogeneous axial resolution across the FOV afforded by the corrected probes could be useful not only to increase the SNR of neural calcium signals, but also to better isolate individual neurons, decreasing signal contamination across neighboring cells. To test this hypothesis, we considered correlations in 'adjacent' simulated neurons, defined as pairs of neurons for which the distance between their centroids was ≤25 μm (the radius of the soma of excitatory neurons in the olfactory cortex is around 10 μm, *Suzuki and Bekkers, 2011*). Compared to uncorrected probes, we found that corrected microendoscopes of both lengths led to lower fraction of pairs of adjacent cells whose activity correlated significantly more than expected (*Figure 6A and F* and *Supplementary file 5*), confirming reduced cross-contamination among cells. The fraction of pairs of adjacent cells (out of the total number of adjacent pairs) whose activity correlated significantly more than expected increased as a function of the peak SNR threshold for corrected and uncorrected microendoscopes of both lengths (*Figure 6A and F*). This effect was due to a larger decrease of the total number of pairs of adjacent cells as a function of the peak SNR threshold compared to the decrease in the number of pairs of adjacent cells whose activity was more correlated than expected (*Figure 6—figure*

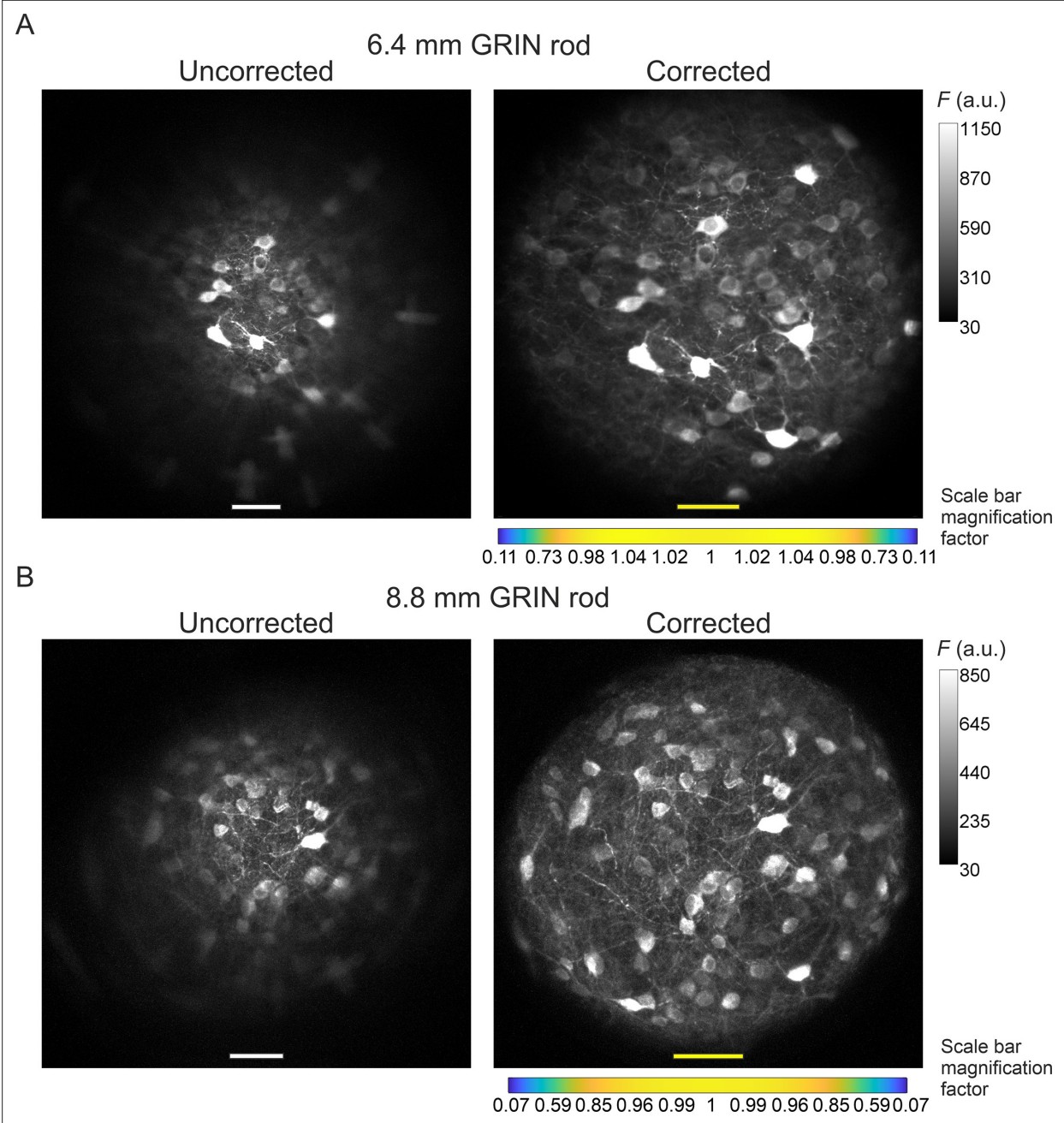

**Figure 4.** Aberration correction in long GRIN lens-based microendoscopes enables high-resolution imaging of biological structures over enlarged FOVs. (**A**) jGCaMP7f-stained neurons in a mouse fixed brain slice were imaged using 2PLSM ($\lambda_{exc}$=920 nm) through an uncorrected (left) and a corrected (right) microendoscope based on the 6.4 mm-long GRIN rod. Images are maximum fluorescence intensity (*F*) projections of a z-stack acquired with a 5 µm step size. Number of steps: 32 and 29 for uncorrected and corrected microendoscope, respectively. Scale bars: 50 µm. Left: the scale applies to the entire FOV. Right: the scale bar refers only to the center of the FOV; off-axis scale bar at any radial distance (*x* and *y* axes) is locally determined multiplying the length of the drawn scale bar on-axis by the corresponding normalized magnification factor shown in the horizontal color-coded bar placed below the image (see also *Figure 3*, *Supplementary file 3*, and Materials and methods for more details). (**B**) Same results for the microendoscope based on the 8.8 mm-long GRIN rod. Number of steps: 23 and 31 for uncorrected and corrected microendoscope, respectively.

The online version of this article includes the following figure supplement(s) for figure 4:

**Figure supplement 1.** Modified version of *Figure 4* with the FOVs of corrected microendoscopes rescaled to match the real pixel size of the FOVs of uncorrected microendoscopes in the center of the image.

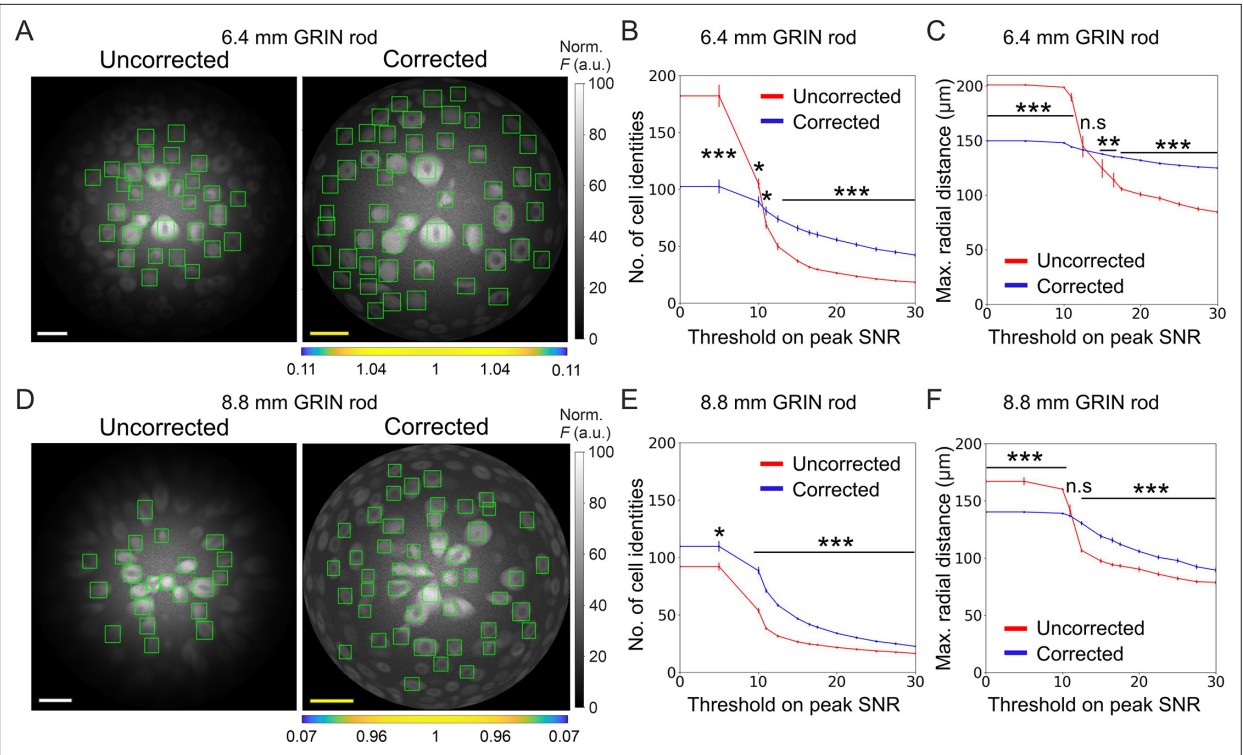

**Figure 5.** Long corrected microendoscopes sample more homogeneously simulated neuronal activity across the FOV. (**A**) Median fluorescence intensity (*F*) projections of representative synthetic t-series for the uncorrected (left) and the corrected (right) 6.4 mm-long microendoscope. Cell identities were detected using CITE-ON *Sità et al., 2022* in n=13 simulated t-series for both the uncorrected and corrected microendoscopes and cellular activity traces were extracted. Green rectangular boxes mark cell identities that have peak SNR of the activity trace higher than a threshold set to peak SNR = 15. Scale bars: 50 μm. Left: the scale applies to the entire FOV. Right: the scale bar refers to the center of the FOV; off-axis scale bar at any radial distance (*x* and *y* axes) is locally determined multiplying the length of the drawn scale bar by the corresponding normalized magnification factor shown in the horizontal color-coded bar placed below the image. (**B**) Number of detected cell identities in simulated FOV as a function of peak SNR threshold imposed on cellular activity traces. Data are mean values ± s.e.m. for both the uncorrected (red) or corrected (blue) case. Statistical significance is assessed with Mann-Whitney U test; *, p<0.05; ***, p<0.001. (**C**) Maximal distance from the center of the FOV at which a cell is detected as a function of peak SNR threshold. Data are mean values ± s.e.m. for both the uncorrected (red) and corrected (blue) case. Statistical significance is assessed with Mann-Whitney U test; **, p<0.01; ***, p<0.001; n.s., not significant. (**D**) Same as (**A**) for the 8.8 mm long microendoscope. (**E,F**) Same as (**B,C**) for n=15 simulated t-series for both the uncorrected and corrected 8.8 mm-long microendoscope.

The online version of this article includes the following source data for figure 5:

**Source data 1.** Numerical values to reproduce graphs in *Figure 5B, C, E and F*.

supplement 1). Moreover, in uncorrected microendoscopes the pairwise correlation of adjacent cells showed a larger linear dependence on the radial distance of the pair from the center of the FOV compared to corrected microendoscopes (*Figure 6—figure supplement 2A* and C, see also Materials and methods). We interpreted these findings on correlated neuronal activity as due to the enlarged PSF of uncorrected microendoscopes in lateral portion of the FOV (*Figure 3G–J*), leading to increased signal contamination by sampling across neurons. These results thus suggest that corrected microendoscopes decrease the artefactual correlation of nearby neurons by diminishing the spatial extension of the excitation volume used for sampling neuronal activity (PSF, *Figure 3G–J*).

Using synthetic data, we could precisely quantify the extent to which correcting aberration in microendoscopes decreased mixing signals across neurons. For each ROI, we computed a general linear model (GLM) of the contributions to fluorescence signals of the activity of all ground truth simulated neurons, called 'sources', (*Figure 6B, C, G and H*). From the GLM coefficients, we extracted a source 'purity' index, which reached its highest possible value of 1 when only one source contributed to the calcium fluorescent trace in the considered ROI and had values <1 when different source neurons were mixed in the fluorescent signal of the considered ROI. For uncorrected microendoscopes of both lengths, we found that purity rapidly decreased with the radial distance of the ROI from the

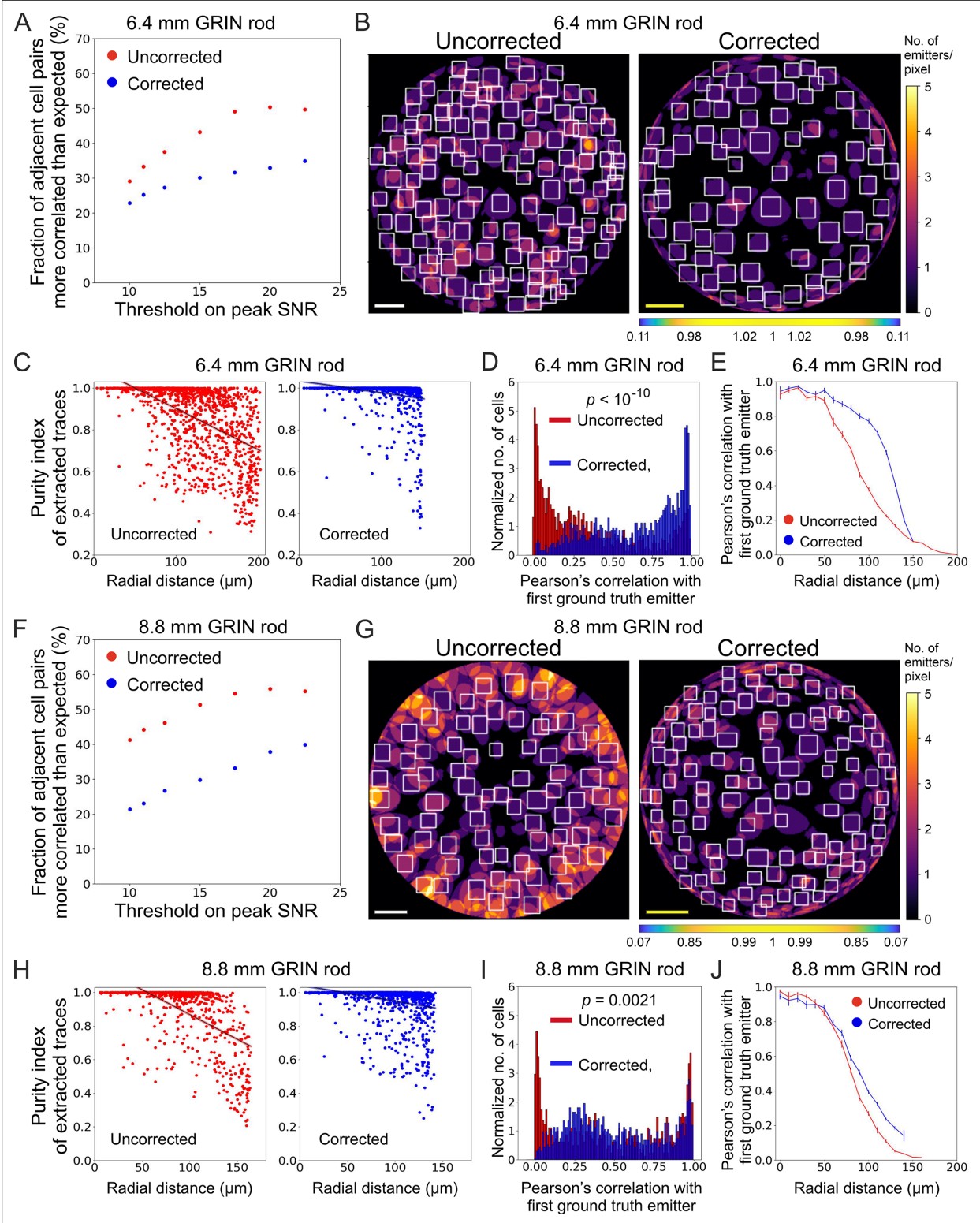

**Figure 6.** Long corrected microendoscopes enable more precise collection of simulated activity signals from individual cellular sources and decrease cross-contamination between adjacent cells. (**A**) Fraction of adjacent cell pairs (distance between detected cell centroids ≤25 μm) that are more correlated that expected (expected pair correlation was estimated as mean Pearson's correlation between ground truth activity traces of any possible neuronal pairs plus 3 SDs) as a function of the peak SNR threshold imposed on extracted activity traces for n=13 simulated experiments with the 6.4 mm-long uncorrected (red) and corrected (blue) microendoscope. (**B**) 2D projection of the intersection between the 3D FOV and the 3D ground truth distribution of light sources for a representative uncorrected (left) and corrected (right) synthetic t-series obtained with a 6.4 mm-long microendoscope.

*Figure 6 continued*

The color scale shows the number of overlapping sources that are projected on the same pixel. White boxes mark cell identities detected using CITE-ON (*Sità et al., 2022*) without any threshold on the peak SNR of activity traces. Scale bars: 50 µm. Left: the scale applies to the entire FOV. Right: the scale bar refers to the center of the FOV; off-axis scale bar at any radial distance (*x* and *y* axes) is locally determined multiplying the length of the scale bar by the corresponding normalized magnification factor shown in the horizontal color-coded bar placed below the image. (**C**) Purity index of extracted traces with peak SNR >10 was estimated using a GLM of ground truth source contributions and plotted as a function of the radial distance of cell identities from the center of the FOV for n=13 simulated experiments with the 6.4 mm-long uncorrected (red) and corrected (blue) microendoscope. Black lines represent the linear regression of data ± 95% confidence intervals (shaded colored areas). Slopes ± s.e.: uncorrected, (–0.0020±0.0002) µm$^{-1}$; corrected, (–0.0006±0.0001) µm$^{-1}$. Uncorrected, n=1365; corrected, n=1,156. Statistical comparison of slopes, p<10$^{-10}$, permutation test. Linear regression was repeated using the same range of radial distances for the uncorrected and corrected case, with the maximum value on the *x*-axis corresponding to the minimum value between the two maximum radial distances obtained in the uncorrected and corrected case (maximum radial distance: 151.6 µm); slopes ± s.e.: uncorrected, (–0.0015±0.0002) µm$^{-1}$; corrected, (–0.0006±0.0001) µm$^{-1}$. Uncorrected, n=991; corrected, n=1,156. Statistical comparison of slopes, p<10$^{-10}$, permutation test. (**D**) Distribution of the Pearson's correlation value of extracted activity traces with the first (most correlated) ground truth source for n=13 simulated experiments with the 6.4 mm-long uncorrected (red) and corrected (blue) microendoscope. Median values: uncorrected, 0.25; corrected, 0.73; the p is computed using the Mann-Whitney U test. (**E**) Pearson's correlation ± s.e.m. of extracted activity traces with the first (most correlated) ground truth emitter as a function of the radial distance for n=13 simulated experiments with the 6.4 mm-long uncorrected (red) and corrected (blue) microendoscope. (**F–J**) Same as (**A–E**) for n=15 simulated experiments with the 8.8 mm-long uncorrected and corrected microendoscope. (**H**) Slopes ± s.e.: uncorrected, (–0.0031±0.0003) µm$^{-1}$; corrected, (–0.0010±0.0002) µm$^{-1}$. Uncorrected, n=808; corrected, n=1,328. Statistical comparison of slopes, p<10$^{-10}$, permutation test. Linear regression using the same range of radial distances for the uncorrected and corrected case (maximum radial distance: 142.1 µm), slopes ± s.e.: uncorrected, (–0.0014±0.0003) µm$^{-1}$; corrected, (–0.0010±0.0002) µm$^{-1}$. Uncorrected, n=718; corrected, n=1328. Statistical comparison of slopes, p=0.0082, permutation test. (**I**) Median values: uncorrected, 0.43; corrected, 0.46; the p is computed using the Mann-Whitney U test.

The online version of this article includes the following source data and figure supplement(s) for figure 6:

**Source data 1.** Numerical values to reproduce graphs in *Figure 6A and C–E*.

**Source data 2.** Numerical values to reproduce graphs in *Figure 6F and H–J*.

**Figure supplement 1.** The total number of detected adjacent cell pairs decreases faster with peak SNR threshold than the number of adjacent cell pairs more correlated than expected does.

**Figure supplement 2.** Aberration correction enables more accurate measurement of population activity.

center of the FOV. In contrast, for corrected microendoscopes the purity index decreased far more slowly with the radial distance and the slope of the linear fit of the purity distribution as a function of the radial distance was significantly smaller than that of the uncorrected probe (*Figure 6C and H*). For any value of the peak SNR threshold used to select neurons, the purity index was also larger for corrected than for uncorrected probes (*Figure 6—figure supplement 2B* and D). Moreover, for each ROI we analyzed the correlation of the extracted traces with the most correlated ground truth neuron ('first ground truth source') as a function of the radial distance of the ROI (*Figure 6D, E,I and J*). For microendoscopes of both lengths, we observed that ROIs from t-series generated with corrected microendoscopes displayed larger correlation with the ground truth signal of an individual neuron over longer distances from the optical axis, compared to uncorrected probes (*Figure 6E and J*). *Figure 6J* showed a moderate enlargement of the distance at which the Pearson's correlation with the first ground truth emitter started to drop. This may appear at first in contrast with the enlargement of the FOV shown in *Figure 4B*. However, it can be understood considering that in *Figure 4B*, all illuminated neurons were visible regardless of whether they were imaged with high axial resolution (e.g. <10 µm as defined in *Figure 3J*) or poor axial resolution. In contrast, in *Figure 6J* we evaluated the correlation between the calcium trace extracted from a given ROI and the real activity trace of the first simulated ground truth emitter for that specific ROI. The moderate increase in the correlation for the corrected microendoscope compared to the uncorrected microendoscope (*Figure 6J*) was consistent with the moderate improvement in the axial resolution of the corrected probe compared to the uncorrected probe at intermediate radial distances (60–100 µm from the optical axis, see *Figure 3J*). Taken together, these results indicated that corrected microendoscopes enabled more precise recording of individual cellular source neurons and enhanced demixing of signals coming from different neurons.

## Validation of long corrected microendoscopes for large FOV population imaging in ventral brain region of awake mice

We tested long corrected microendoscopes for in vivo 2P calcium imaging experiments in ventral regions of the mouse brain. To this aim, we implanted mice with corrected microendoscopes based on 8.8 mm-long GRIN lenses over the mouse anterior olfactory cortex, a deep region laying >4 mm down the mouse brain surface (*Figure 7A and B*). Mice expressed genetically encoded calcium indicators, either jGCaMP7f or jGCaMP8f, in excitatory neurons through adeno-associated virus (AAVs) injection. In awake head-fixed mice, we recorded 2P imaging t-series ($\lambda_{exc}$=920 nm; frame rate: 30 Hz; t-series duration: 8 min) during repetitive olfactory stimulation with various odorants (*Figure 7C–E*, see Materials and methods).

In recorded t-series, we first observed that the peak SNR of calcium events recorded in individual cells was constant across the FOV (*Figure 8A*). We then computed the pairwise correlation of the activity traces of pairs of adjacent neurons (defined as pairs of cells with an intersomatic distance of ≤25 μm) as a function of the radial distance of the pair's position centroid from the center of the FOV (*Figure 8B*). We found no dependence of the pairwise correlation on the distance from the center of the FOV. In fact, the slope of the linear fit of pairwise correlation as a function of the radial distance was not significantly different from zero (*Figure 8B*). These findings were in agreement with the results of the optical characterization (*Figure 3*) and with those of the simulation of calcium data (*Figures 5 and 6*). These results suggested that the absence of dependence of the pairwise correlation with radial distance reflected the improved and more homogeneous spatial resolution of corrected microendoscopes, which prevented source mixing and artificially inflated correlations of adjacent neurons in the FOV periphery. Varying the distance between centroids of adjacent neurons did not change the result (*Supplementary file 6*). Running the analysis separately for mice expressing jGCaMP8f and jGCaMP7f also confirmed the previous findings (*Supplementary file 6*). Finally, the Pearson's correlation coefficient between the activity traces of any pair of neurons within the FOV decresed as a function of their reciprocal distance (*Figure 8C*). This decrease in correlation could reflect a slight change in the laminar position of recorded neurons or suggest a decrease in recurrent connectivity between neurons of the olfactory cortex with distance (*Franks et al., 2011*; *Hagiwara et al., 2012*).

Combined with the results of the simulated calcium data, these in vivo findings demonstrate that, by decreasing off-axis aberrations of commercial GRIN lenses, long corrected microendoscopes enable unbiased sampling of neural population activity on a large FOV in ventral regions of the mouse brain in vivo.

## Discussion

Combining GRIN lens-based endoscopy (*Jung and Schnitzer, 2003*; *Jung et al., 2004*) with 2P fluorescence imaging (*Helmchen and Denk, 2005*; *Denk et al., 1990*) is a powerful approach to study the physiology of deep brain circuits. However, GRIN lenses are not ideal optical elements and they have intrinsic optical aberrations (*Bortoletto et al., 2011*; *Lee and Yun, 2011*), which significantly degrade the PSF and reduce the effective FOV (*Antonini et al., 2020*; *Wang and Ji, 2012*; *Wang and Ji, 2013*; *Li et al., 2024*). In the current study, we developed, characterized, and validated new aberration corrected microendoscopic probes to address this critical need. Corrected microendoscopes had length of 6.4 mm and 8.8 mm (see *Table 2* and Materials and methods), allowing reaching the deepest structures of the mouse brain, and cross section of 500 μm, ensuring limited tissue invasiveness.

We first designed the corrected microendoscopes using optical ray-race simulations. Simulations showed strongly aberrated PSF in lateral portions of the FOV of uncorrected microendoscopes and predicted that single corrective optical elements with the aspherical profiles shown in *Figure 1A and B* improved the optical performance of microendoscopes of either length. This prediction was experimentally confirmed using subresolved fluorescent beads and layers. We first observed that uncorrected microendoscopes of either length displayed strong astigmatism and had largely distorted PSF in lateral regions of the FOV. In corrected microendoscopes, we instead found that the PSF had regular shape and constrained values for larger radial distances, leading to more homogeneous spatial resolution across the FOV and enlarged effective FOV. Importantly, experimental values describing the improvement of the optical performance of the corrected microendoscopes (*Figure 3G–J* and *Supplementary file 4*) were generally in good agreement with the prediction of the optical simulations

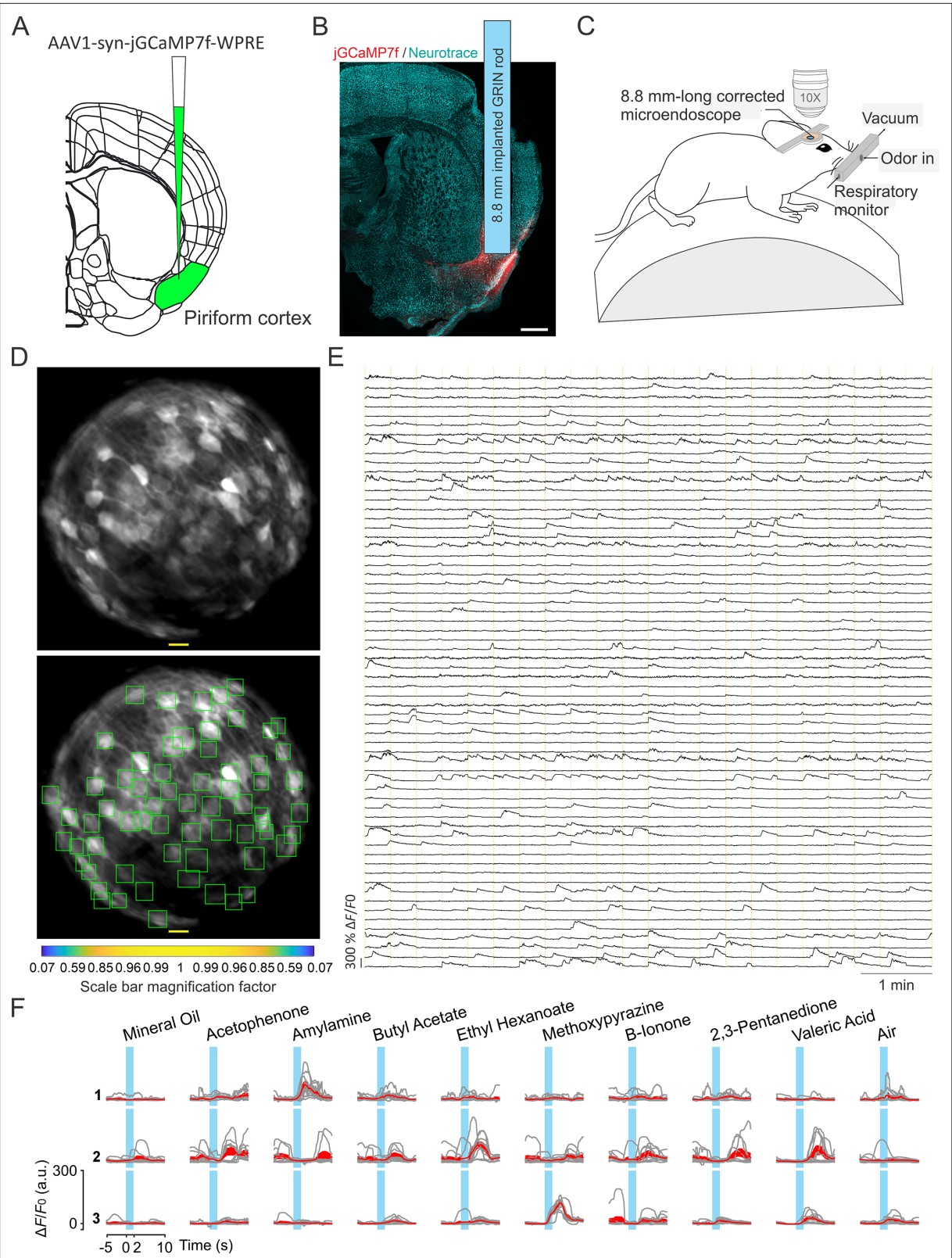

**Figure 7.** Enlarged FOV population imaging of ventral regions of the brain with long corrected microendoscopes in awake mice. (**A**) Schematic showing injection of the viral solution in the mouse piriform cortex. (**B**) Representative section of fixed brain tissue showing the position of the GRIN lens implant. jGCaMP7f fluorescence is shown in red. NeuroTrace (Neurotrace) Nissl staining is shown in cyan. (**C**) Schematic showing the experimental preparation for 2P corrected microendoscope imaging in awake mice. (**D**) Top and bottom: example of FOV with excitatory neurons expressing the calcium sensor

*Figure 7 continued on next page*

*Figure 7 continued*

jGCaMP7f, obtained using the 8.8 mm-long corrected microendoscope in the mouse piriform cortex. The bottom image shows the rectangular boxes (green) indicating the position of detected neurons generated by CITE-ON (see Materials and methods, *Sità et al., 2022*). On-axis scale bar: 20 µm; for off-axis scale bar, refer to the magnification factor bar below the image (see Materials and methods). (**E**) Representative jGCaMP7f activity traces extracted from the recording shown in (**D**) through 8-min-long continuous 2P imaging recordings ($\lambda_{exc}$=920 nm) in an awake head-fixed mouse. Vertical dashed orange lines mark the onset and the end of the 22-s-long trials in which the imaging session was divided. (**F**) Examples of neuronal responses to different olfactory stimuli for three representative cells. For each odor (top labels), gray lines represent calcium responses recorded in single trials and the red line represent the average response. Light blue areas indicate the time interval of stimulus presentation.

(*Figure 1D and F* and *Supplementary file 2*). The size of simulated PSFs at a given radial distance (e.g. 90 µm, *Figure 1*) tended to be generally smaller than that of the experimentally measured PSFs (*Figure 3*). This might be due to multiple reasons. First, simulated PSFs were excitation PSFs, that is they described the intensity spatial distribution of focused excitation light. On the contrary, measured PSFs resulted from the excitation and emission processes and thus they were also affected by aberrations of light emitted by fluorescent beads and collected by the microscope. Second, in the optical simulations first-order and not higher-order aberrations were considered. Third, intrinsic variability in experimental parameters (e.g. alignment of the microendoscope to the optical axis of the microscope, distance between the GRIN back end and the objective) were not considered in the simulations. It is important to note that the improvement in spatial resolution of GRIN lens-based probes described in this work was related exclusively to the introduction of the corrective lens. In fact, uncorrected microendoscopes used for comparison consisted of a GRIN rod and a glass coverslip attached to the back end of the GRIN lens. Previous work showed that the addition of a glass coverslip, that typically introduces negative spherical aberrations, counteracted positive spherical aberrations introduced by GRIN lens singlets and improved the optical performance of the GRIN rod when combined with the use of a microscope objective with corrective collar (*Murray and Levene, 2012*). Therefore, it is reasonable to speculate that adopting our corrected microendoscopes rather than using bare GRIN lenses enhances imaging performance even more than what described in the present study.

Did the improved optical properties of corrected microendoscopes increase the quality of imaging when corrected microendoscopes were applied to biological samples? We directly addressed this question by performing 2P imaging with uncorrected and corrected microendoscopes in brain slices in which neurons expressed a green fluorescent indicator. In uncorrected microendoscopes, we found that neuronal cell bodies were bright and well contrasted only in the center of the FOV and that neuronal cell bodies were dimmer and blurred in lateral regions of the FOV (*Figure 4*). Moreover, neuronal processes could be observed only in the center of the FOV and not in lateral portions of the FOV (*Figure 4*). In contrast, in corrected microendoscopes of either length, we found that neurons were bright and well contrasted across the whole FOV, which enabled distinguishing neurons, which were blurred and merged together when imaged with uncorrected microendoscopes (*Figure 4*). Furthermore, small neuronal processes could be observed throughout the entire FOV. These imaging findings in brain slices are in line with the results of the optical simulation and of the experimental characterization, which showed improved PSF and more homogeneous spatial resolution in corrected microendoscopes compared to uncorrected ones. It must be considered, however, that the extended FOV achieved by our aberration correction method was characterized by a curved focal plane. Therefore, cells located in different radial positions within the image were located at different axial positions and cells at the border of the FOV were closer to the front end of the microendoscope.

How does the improved optical performance of corrected microendoscope translate into enhanced ability to extract meaningful information from population of neurons in functional fluorescence imaging experiments? We addressed this question generating and analyzing synthetic calcium data in which we could compare the signals extracted from simulated recordings using uncorrected and corrected microendoscopes with the ground truth signals, which were used to generate the simulations. We found that the aberration correction led to an increase in the number of ROIs with high SNR and, correspondingly, a decrease in the number of ROIs with low SNR. Moreover, aberration correction led to high and more stable purity of source demixing over radial distance, which, in turn, led to a major decrease in the artificial overestimation of correlations between adjacent neurons due to source mixing. Since investigating neuronal population codes needs quantification of neuronal activity with large SNR, high precision, and limited contamination, we expect the corrected microendoscopes

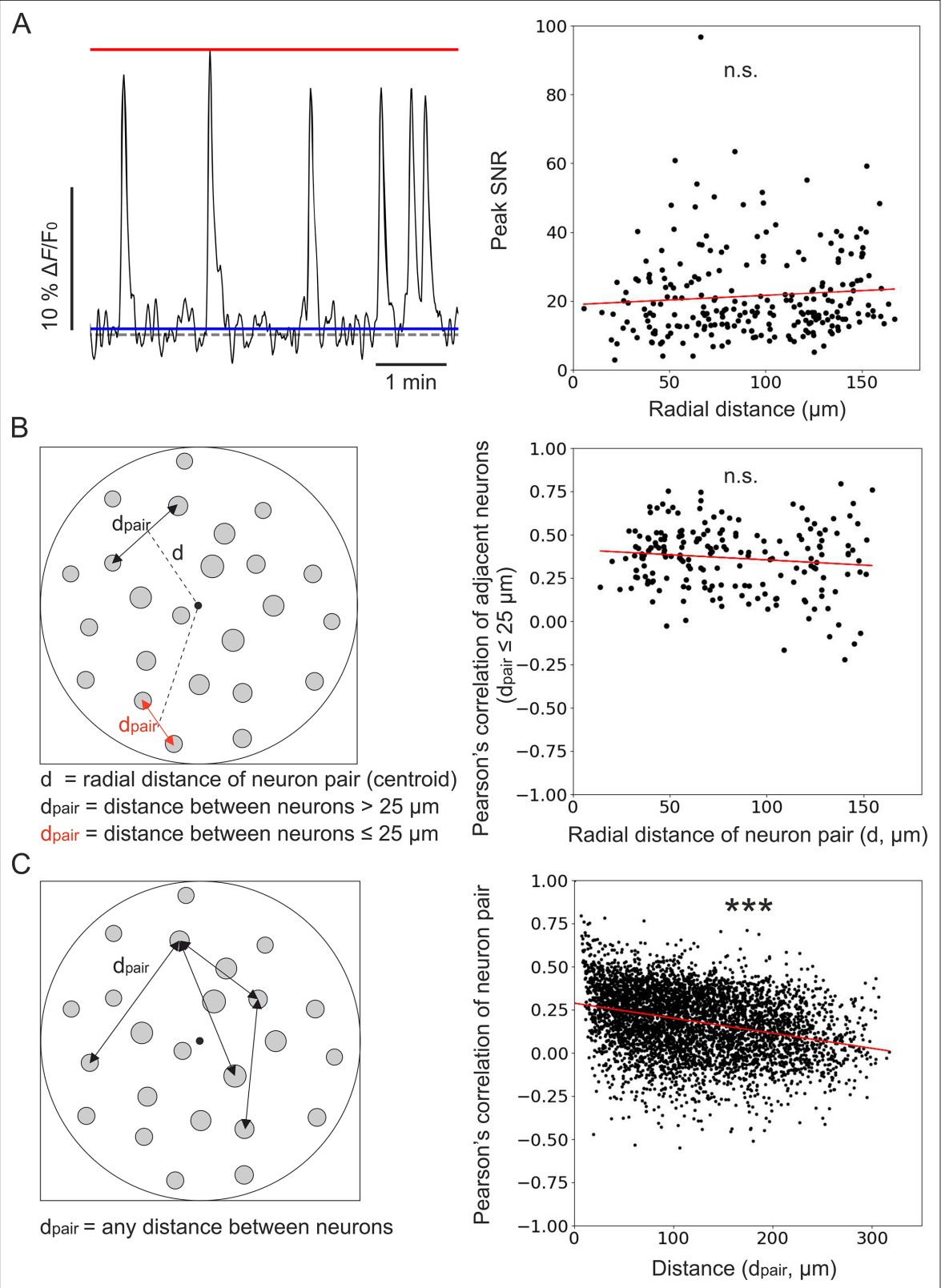

**Figure 8.** Unbiased population imaging in the piriform cortex of awake mice using long corrected microendoscopes. (**A**) Left: schematic representation of trace parameters used to compute peak SNR. The red solid line marks the maximum value of the trace, the blue line indicates the SD of intensity values below the 25th percentile of the intensity distribution of the entire trace (see Materials and methods), and the dashed grey line marks the origin of the $y$-axis ($\Delta F/F_0$=0). Right: peak SNR of calcium traces extracted from individual cells as a function of radial distance of the cell (n=240 neurons from

*Figure 8 continued on next page*

*Figure 8 continued*

m=6 FOVs). The red line is the linear regression of data points: intercept ±s.e.=19 ± 2; slope ±s.e. = (0.03±0.02) µm⁻¹. The slope is not significantly different from zero, p=0.18, permutation test. (**B**) Left: schematic representation showing pairs of neurons and distance definitions for the Pearson's correlation analysis shown on the right (in red, distance between 'adjacent neurons' $d_{pair}$ ≤25 µm). Right: Pearson's correlation of calcium traces from 'adjacent neurons' as a function of radial distance of the pair centroid from the center of the FOV (n=195 adjacent neuron pairs from m=6 FOVs). The red line represents the linear regression of data points: intercept ±s.e.=0.41 ± 0.03; slope ±s.e. = (–0.0006±0.0004) µm⁻¹. The slope is not significantly different from zero, p=0.089, Wald test. (**C**) Left: schematic representation of neuron pairs used for the analysis on the right (any possible $d_{pair}$). Right: Pearson's correlation of calcium traces from pairs of neurons as a function of the distance between them (n=4767 pairs from m=6 FOVs). The red line is the linear regression of data points: intercept ±s.e.=0.288 ± 0.005; slope ±s.e. = (–87±4) · 10⁻⁵ µm⁻¹. The slope is significantly different from zero, p=2 · 10⁻⁵, permutation test.

The online version of this article includes the following source data for figure 8:

**Source data 1.** Numerical values to reproduce graphs in *Figure 8A–C*, right panels.

presented in this manuscript will greatly contribute to the investigation of deep brain regions. In particular, the reduction of artificial overestimation of correlations due to source mixing appears crucial to precisely evaluate within-network neuron-to-neuron communication and investigate the role of correlations between neurons in population codes. Artificial correlations between nearby neurons would bias estimates of neuron-to-neuron communication toward overestimating communications at short distances. Moreover, because source mixing duplicates one neuron into another and thus can only increase redundancy, artificial correlations would also bias the effect of correlations on population coding by artificially inflating redundant correlations over synergistic interactions between neurons (*Ecker et al., 2010*; *Panzeri et al., 2022*). Thus, correcting optical aberrations in GRIN lenses as presented in this study appears essential for the unbiased characterization of single-neuron and population codes in deeper brain structures.

We used long corrected microendoscopes to measure population dynamics in the olfactory cortex of awake head-restrained mice with unprecedented combination of high spatial resolution across the FOV and minimal invasiveness (*Wang et al., 2020*). In agreement with the results of our optical characterization and of the synthetic calcium data, we found that the SNR of calcium signals was independent on the radial distance and that high SNR calcium signals could be recorded across the whole FOV. Moreover, we observed that the correlation between calcium signals of adjacent neurons did not depend on the radial distance, in line with the more homogeneous PSF that characterizes corrected microendoscopes. The correlation between the calcium signals of any pair of neurons displayed, instead, a negative dependence on the distance between the two neurons. This decrease in correlation could reflect a slight change in the laminar position of recorded neurons or suggest a decrease in recurrent connectivity between neurons of the olfactory cortex with distance (*Franks et al., 2011*; *Hagiwara et al., 2012*). In relationship to studies on the olfactory cortex, the improved optical characteristics of corrected microendoscopes increase our ability to understand the properties of this network in several fundamental ways. First, imaging more cells with increased SNR improves our ability to decipher information encoded by neural ensembles in the olfactory cortex. For example, we will be able to characterize the odor response properties of genetically defined subpopulations of olfactory cortex neurons, which may be too

**Table 2.** Characteristics of commercially available (Commercial) and customized (Custom) GRIN rods used for simulation and fabrication of aberration corrected microendoscopes.
All GRIN rods were obtained from GRINTECH GmbH, Jena, Germany.

| | 6.4 mm-long GRIN rod | | 8.8 mm-long GRIN rod | |
|---|---|---|---|---|
| | Commercial | Custom | Commercial | Custom |
| Catalogue # | NEM-050-25-10-860-S-1.5p | NEM-050-25-15-860-S-1.5p | NEM-050-25-10-860-S-2.0p | NEM-050-23-15-860-S-2.0p |
| Diameter (µm) | 500 | 500 | 500 | 500 |
| Length (mm) | 6.52 | 6.4 | 8.85 | 8.8 |
| Working distance (µm) | Image side: 100 Object side: 250 | Image side: 150 Object side: 250 | Image side: 100 Object side: 250 | Image side: 150 Object side: 230 |

sparse to be reliably captured in the smaller effective FOV of uncorrected GRIN lenses. Second, imaging more cells with high SNR in each mouse will allow us to explore differences between individual mice, rather than pooling cells from different animals. Third, less cross-contamination of fluorescence signals improves cell segmentation and motion correction, which are major challenges when imaging mice that are awake or actively engaged in behavioral tasks (*Harris et al., 2016*; *Brondi et al., 2020*). Finally, improved cell segmentation facilitates cell tracking in chronic imaging experiments. Thus, corrected microendoscopes will enhance our ability to monitor individual cells over extended periods of time, for example while mice learn to perform an odor discrimination task (*Berners-Lee et al., 2023*).

Alternative methods to correct optical aberrations in GRIN lenses include, for example, low-order adaptive optics (*Bortoletto et al., 2011*), adaptive optics through pupil segmentation (*Wang and Ji, 2012*; *Wang and Ji, 2013*), and geometric transformation adaptive optics (*Li et al., 2024*). In the first approach, an adaptive optics module comprising an electrostatic membrane mirror was introduced in the optical path of the 2P microscope and it was shown to significantly improve the microscope PSF (*Bortoletto et al., 2011*). In adaptive optics through pupil segmentation instead, the objective rear lens was divided into N different subsections. By comparing images acquired with light going through each individual pupil segment with images acquired with light going through the whole pupil, local wavefront tilts introduced by aberrations were calculated and corrected for using a spatial light modulator (SLM; *Ji et al., 2010*). This method was applied to GRIN lens-based endoscopes of larger diameter (i.e. 1.4 mm) compared to the one used in this study (i.e. 0.5 mm) and it was shown to enlarge the effective FOV during 2P imaging (*Wang and Ji, 2012*; *Wang and Ji, 2013*). Finally, in geometric transformation adaptive optics, a 90 degrees rotation was applied to the aberrated beam spatial profile exiting a first GRIN lens and the aberrated beam was passed through a second GRIN lens. This strategy was effective in correcting chromatic and volumetric aberrations in GRIN microendoscopes of small cross section (500 µm *Li et al., 2024*). However, all adaptive optics methods described above require substantial change the optical set-up (e.g. the insertion of an SLM, membrane mirror, and additional optics) and the development of specific software control (*Bortoletto et al., 2011*; *Wang and Ji, 2012*; *Wang and Ji, 2013*; *Li et al., 2024*). Additionally, under certain conditions pupil segmentation may limit the acquisition frame rate of the system (*Wang and Ji, 2013*). In contrast to the methods described above, the long corrected microendoscopes presented in the current manuscript do not limit the temporal resolution of the acquisition system, require no modification of the hardware and software of the optical set-up, and are easily coupled to any standard 2P microscope. Another advantage of long corrected microendoscopes described here over adaptive optics approaches is the possibility to couple corrected microendoscopes with portable 2P microscopes (*Zong et al., 2017*; *Zong et al., 2021*; *Zong et al., 2022*), allowing high-resolution functional imaging of deep brain circuits on an enlarged FOV during naturalistic behavior in freely moving mice. The corrected microendoscopes presented in this study thus represent a ready-to-use solution, which will likely ease the adoption of aberration correction in GRIN lens-based 2P functional imaging across experimental laboratories.

In conclusion, we designed, developed, and validated a new series of aberration-corrected GRIN lens-based microendoscopes, which are long enough to reach the most ventral regions of the mouse brain. Long corrected microendoscopes had improved axial resolution and several folds enlarged effective FOV. Importantly, these optical improvements enabled more precise extraction of single cell and population signals from 2P imaging recordings. We validated long corrected microendoscopes by performing functional imaging of the olfactory cortex circuits in awake head-fixed mice. Long corrected microendoscopes provide a unique combination of high and homogenous optical properties across the FOV and minimal invasiveness. Furthermore, they do not require modification of the hardware and software of the optical microscope, providing a ready-to-use solution to experimental laboratories. We foresee that the long corrected microendoscopes developed in this study will drastically improve the yield of 2P imaging in deep brain areas of mice and larger mammals, such as rats (*Scott et al., 2013*), cats (*Kara and Boyd, 2009*; *Ohki et al., 2005*), and marmosets (*Sadakane et al., 2015*; *Ebina et al., 2018*).

## Materials and methods

**Key resources table**

| Reagent type (species) or resource | Designation | Source or reference | Identifiers | Additional information |
|---|---|---|---|---|
| Strain, strain background (*Mus musculus*; males, females) | C57BL/6 J | Jackson Laboratory | Strain #: 000664; RRID:IMSR_JAX:000664 | |
| Genetic reagent (*Mus musculus*; males, females) | Ai14 | Jackson Laboratory | Strain #: 007914; RRID:IMSR_JAX:007914 | |
| Recombinant DNA reagent | pGP-AAV-syn-jGCaMP8f-WPRE | Addgene, *Zhang et al., 2023* | Catalog #: 162376-AAV1 | Dilution: 1/3 in PBS |
| Recombinant DNA reagent | AAV1-syn-jGCaMP7f-WPRE | Addgene *Dana et al., 2019* | Catalog #: 104488-AAV1 | Dilution: 1/3 in PBS |
| Chemical compound, drug | Photoresin | Nanoscribe | Resin: IP-S | |
| Chemical compound, drug | UV-curable glue, NOA63 | Norland Products | Product: NOA63 | |
| Chemical compound, drug | Poly-l-lysine | Merck | Product No.: P2636 | Dilution: 0.1 mg/mL in water |
| Commercial assay or kit | PDMS, Sylgard 184 Silicone Elastomer Kit | Dow | Material No.: 4019862 | |
| Commercial assay or kit | Metabond adhesive cement | Parkell | SKU: S396 | |
| Commercial assay or kit | Dental cement, Pi-ku-plast HP 36 Precision Pattern Resin | XPdent | SKU #: 54000210 SKU #: 54000215 | |
| Commercial assay or kit | Kwik-Sil silicone elastomer | World Precision Instruments | Code: KWIK-CAST | |
| Software, algorithm | OpticStudio 15 | Zemax (Ansys) | | https://www.ansys.com/products/optics/ansys-zemax-opticstudio |
| Software, algorithm | CAD | SolidWorks | | https://www.solidworks.com/product/solidworks-3d-cad |
| Software, algorithm | Imagej/Fiji | NIH (open source) | RRID:SCR_002285 | https://fiji.sc/ |
| Software, algorithm | Matlab R2022b, Matlab | MathWorks | RRID:SCR_001622 | https://it.mathworks.com/products/new_products/release2022b.html |
| Software, algorithm | CITE-ON | *Sità et al., 2022*; *Sità et al., 2021* | | Optical Approaches to Brain Function Laboratory, Istituto Italiano di Tecnologia; https://gitlab.iit.it/fellin-public/cite-on |
| Software, algorithm | Python 3.7, Python | Python Software Foundation | RRID:SCR_008394 | https://www.python.org/downloads/release/python-370/ |
| Software, algorithm | Software for generation of artificial t-series | This paper; *Antonini et al., 2020* | | Optical Approaches to Brain Function Laboratory, Istituto Italiano di Tecnologia. See Data and software availability section in this paper and in *Antonini et al., 2020* |
| Other | 6.4 mm-long GRIN rod | GRINTECH | Product code: NEM-050-25-10-860-S-1.5p | Customized product |
| Other | 8.8 mm-long GRIN rod | GRINTECH | Product code: NEM-050-25-10-860-S-2.0p | Customized product |
| Other | 3D microprinter | Nanoscribe | Photonic Professional GT2+ | |
| Other | Chameleon Ultra II laser source | Coherent | Chameleon Ultra II | |
| Other | Chameleon Discovery laser source | Coherent | Chameleon Discovery | |

| Reagent type (species) or resource | Designation | Source or reference | Identifiers | Additional information |
|---|---|---|---|---|
| Other | Investigator scan head | Bruker Corporation | Ultima Investigator | |
| Other | Ultima IV scan head | Bruker Corporation | Prairie Technologies Ultima IV | |
| Other | 20 x dry objective, objective | Zeiss | EC Epiplan-Neofluar 20 x/0,50 Ph2 M27 | For optical characterizations |
| Other | 10 X Plan Apochromat Lambda objective, objective | Nikon | Plan Apochromat Lambda D 10 x/0.45 | For in vivo imaging |
| Other | 16-channel olfactometer | AutoMate Scientific | | Customized product |
| Other | Photoionization detector | Aurora Scientific | Product: 200B: miniPID Fast Response Olfaction Sensor | |
| Other | Calibration ruler, 4-dot calibration slide | Motic | Product code: 1101002300142 | |
| Other | Fluorescent beads, FluoSpheres Carboxylate-Modified Microspheres | Thermo Fisher | Catalog No.: F8803 | Diameter: 100 nm |
| Other | Teensy 3.6 | PJRC | Product: Teensy 3.6 Development Board | |

## Ray-trace simulations of corrected microendoscopes

To identify the design of the corrective optics to obtain aberration corrected microendoscopes, we performed ray-trace simulations using OpticStudio 15 (Zemax, Kirkland, WA, US) and integrating the optical model of two commercially available GRIN rods (NEM-050-25-10-860-S-1.5p and NEM-050-25-10-860-S-2.0p, GRIN needle endomicroscopes, GRINTECH Gmbh, Jena, Germany). The wavelength selected for simulations was 920 nm, the commonly used excitation wavelength for 2P imaging of green-emitting fluorescent indicators (*Chen et al., 2013*; *Dana et al., 2019*). In simulations, we placed the aspheric lens, whose refractive index corresponded to that of the UV-curable glue we used for microendoscope fabrication (NOA63, see also below), on one side and the back end of the GRIN rod on the other side. 100 μm of BK7 glass were interposed between the corrective lens and the GRIN rod. The profile of the aspheric lens was identified by optimizing a merit function, which took into account the Strehl ratio of focal spots simulated at different field radial distances (up to 200 μm for the 6.4 mm-long and up to 170 μm for the 8.8 mm-long GRIN rod) after being relayed by the optical assembly. During the optimization, the working distance was constrained to values compatible with neurophysiological experiments (~180–220 μm) and the refractive indices simulated at the two ends of the optical assembly were 1 (air) and 1.33–1.36 (brain tissue). The following formula described the surface profile of aspheric lenses similarly to *Antonini et al., 2020*:

$$Z(r) = \frac{cr^2}{1 + \sqrt{1 - (1+k)\,c^2 r^2}} + \sum_n \alpha_n r^{2n}, \tag{1}$$

where $r$ is the radial distance from the optical axis; $c$ is equal to 1/R, where R is the radius of curvature; $k$ is the conic constant; $\alpha_n$ are asphericity coefficients. We changed the parameters $c$, $k$, $\alpha_n$ (with $n$=1–4) in *Equation 1* to maximize the Strehl ratio (*Smith, 2008*) over the largest possible area of the FOV. We obtained simulated 2P PSFs (*Figure 1*) by 3D sampling the squared calculated Strehl ratio (*Smith, 2008*). To compute the spatial resolution of simulated microendoscopes, we fitted $x,y,z$ intensity profiles of the simulated PSFs with Gaussian curves. The ray-trace simulations demonstrated improved Strehl ratio values with GRIN lenses of slightly longer working distances (*Table 2*, Custom) compared to the two commercial GRIN rods initially considered (*Table 2*, Commercial). We thus used the GRIN lenses of custom working distance indicated in *Table 2* for all results presented in this manuscript. To evaluate the impact of corrective lenses on the optical performance of GRIN-based microendoscopes, we also simulated uncorrected microendoscopes composed of the same optical elements of corrected probes (glass coverslip and GRIN rod), but in the absence of the corrective lens.

**Table 3.** Working distances of uncorrected and corrected microendoscopes based on long GRIN lenses obtained with ray-trace simulations shown in **Figure 1**.

| | Microendoscope based on 6.4 mm-long GRIN rod | | Microendoscope based on 8.8 mm-long GRIN rod | |
|---|---|---|---|---|
| | Uncorrected | Corrected | Uncorrected | Corrected |
| Working distances (μm) | Image side: 150 Object side: 215 | Image side: 150 Object side: 210 | Image side: 145 Object side: 228 | Image side: 143 Object side: 178 |

Final working distances of simulated uncorrected and corrected microendoscopes based on custom GRIN rods are indicated in **Table 3**.

## Aspherical lens microprinting and microendoscope assembly

The aspheric profiles of the corrective lenses obtained with optical simulations were converted into 3D structure files by a CAD software (SolidWorks, Waltham, MA, US). The aspherical lens prototypes were then fabricated with a commercial 3D microprinter based on 2P polymerization (Photonic Professional GT2+, Nanoscribe, Karlsruhe, Germany) using a proprietary photoresin (IP-S, Nanoscribe, Karlsruhe, Germany) and a ×25 0.8 NA microscope objective in Dip-in Laser Lithography (DiLL) configuration. Once the lens' volume was fully polymerized in Solid printing mode, the residual unpolymerized material was removed with immersion in propylene glycol methyl ether acetate (PGMEA) for 10 min, followed by ~5 min immersion in isopropyl alcohol. The lens prototypes were finally coated with 200 nm of gold with a sputter coater and inspected with scanning electron microscopy to confirm absence of surface defects.

To enable fast manufacturing of multiple corrected microendoscopes, we generated replicas of the aspherical lenses using a molding procedure (**Figure 2C**) as in **Schaap and Bellouard, 2013**. We first fabricated negative molds with PDMS (Sylgard 184 Silicone Elastomer Kit, Dow, Midland, MI, US) casted onto the aspherical lens prototype and solidified at room temperature for 48 hr. We then deposited a small drop of optically transparent UV-curable glue (NOA63, Norland Products, Cranbury, NJ, US) to fill the negative mold. Extreme care was used to avoid bubble formation in this experimental step. We placed a round glass coverslip (coverslip diameter: 3 or 5 mm in diameter; coverslip thickness: 100 μm) on top of the drop of UV-curable glue and pressured it against the PDMS mold (**Figure 2C**). NOA63 was solidified with UV light exposure for 10 min. After UV curing, the corrective lens was visually inspected at the stereomicroscope. In case of formation of air bubbles, the corrective lens was discarded (yield of the molding procedure: ~90 %, n>30 molded lenses). The coverslip with the attached corrective lens was sealed to a customized metal or plastic support ring of appropriate diameter (**Figure 2C**). The support ring, the coverslip and the aspherical lens formed the upper part of the corrected microendoscope, to be subsequently coupled to the proper GRIN rod (**Table 2**, Custom) using a custom-built optomechanical stage and NOA63 (**Figure 2C**; **Antonini et al., 2020**). The GRIN rod was positioned perpendicularly to the glass coverslip, on the other side of the coverslip compared to the corrective lens, and aligned to the aspherical lens perimeter (**Figure 2C**) under the guidance of a wide field microscope equipped with a camera. The yield of the assembly procedure for the probes used in this work was 100% (n=27 endoscopes). For further details on the assembly of corrected microendoscope see **Antonini et al., 2020**.

## Optical characterization of corrected microendoscopes

We performed optical characterization of corrected and uncorrected microendoscopes using 2 P laser scanning microscopes equipped with femtosecond tunable laser sources (80 MHz repetition rate, Chameleon Ultra II or Chameleon Discovery, Coherent, Inc, Santa Clara, CA, US) tuned at 920 nm excitation wavelength and coupled to an Ultima IV or an Investigator scan head (Bruker Corporation, Billerica, MA, US). We used a ×20 dry objective (EC Epiplan-Neofluar, NA = 0.5, Zeiss, Oberkochen, Germany) and coupled it with the microendoscopes using a custom optomechanical mount similarly to **Antonini et al., 2020**. The mount was equipped with a micrometric translator to adjust the distance between the objective lens and the upper surface of the corrected microendoscope. We acquired series of z-stacks of different calibration samples (subresolved fluorescent films, subresolved fluorescent beads, calibration rulers, and mouse fixed brain slices) moving the whole mount carrying both

the objective and the microendoscope, so that their reciprocal distance remained fixed during the acquisition. For all measurements, the distance between the objective and the upper surface of the coverslip was set to 100 μm for uncorrected microendoscopes and to 150 μm and 180 μm for the 6.4 mm-long and the 8.8 mm-long corrected microendoscopes, respectively. This choice was made taking into account the working distances of simulated corrected probes (*Table 3*). The front end of the GRIN rod was immersed into a drop of distilled water, or phosphate-buffered saline (PBS) for fixed brain slices, which was placed over the calibration sample. Due to the short working distance of the micro-endoscopes (*Table 3*), we put no coverslip over the samples. For each measurement, experimental replicates were obtained using corrected microendoscopes assembled with distinct optical elements (i.e. different GRIN lenses and different corrective lenses). The uncorrected microendoscopes were assembled either using different optical elements compared to the corrected ones, or were obtained from the corrected probes after the mechanical removal of the corrective lens. The analysis of z-stacks acquired through calibration samples was performed using Imagej/Fiji and custom code written in Matlab (R2022b, MathWorks, Natick, MA, US). In the following paragraphs, we defined 'nominal' pixel size as the one determined by imaging a calibration ruler (see below) with the microscope objective alone.

## Subresolved fluorescent films

To evaluate the axial extension of the excitation volume for the two types of microendoscopes as a function of the radial distance, we imaged subresolved homogeneous fluorescent layers (thickness: ~300 nm) mounted on a microscope slide. We acquired z-stacks at 256 pixels × 256 pixels resolution (nominal pixel size: 1.73 μm/pixel) with 2 μm axial step and 16 frame averaging. To analyze the axial profile of the imaged film, for each z-stack we generated the mean *x,z* intensity projection by aver-aging 180 evenly distributed sections passing through the center of the FOV of the microendoscope. This *x,z* fluorescence intensity projection represents the section of the FOV. From this image, we then extracted the fluorescence intensity profiles along 100 segments evenly distributed across the FOV and orthogonal to the line that is locally tangent to the film profile. We measured the thickness of the average film profile across the FOV as FWHM of Gaussian curves fitting the extracted fluorescence intensity profiles along the segments. Data were binned along the *x* coordinate of the image (bin size: 10 pixels). To set the zero of the *x* coordinate, we fitted the longitudinal intensity profile of the section of the film (along the section of the FOV) to a Gaussian curve and we selected the pixel corresponding to the peak. We finally averaged together binned data obtained from multiple subresolved layer acquisitions using different microendoscopes.

## Calibration rulers

To properly calibrate microendoscope images, we used a calibration ruler with ticks spaced every 10 μm along two orthogonal directions (4-dot calibration slide, Cat. No. 1101002300142, Motic, Hong Kong). The ruler was oriented such that the ruler ticks were along the *x* and *y* directions of the acquired image and the crossing point between the two orthogonal directions of the ruler was posi-tioned at the center of the microendoscope FOV. z-stacks of the ruler were acquired at 512 pixels × 512 pixels resolution (nominal pixel size: 1.06 μm/pixel) with 2 μm axial step and 8 frame averaging. For each z-stack, we generated the maximum intensity projection and identified tick locations as the local maxima of the intensity profiles obtained along two lines drawn along the two ruler directions. We then counted the number of pixels between the identified tick locations and the center of the FOV. Due to the circular symmetry of GRIN lens-based microendoscopes, we averaged together data obtained along the four radii of the ruler. We finally averaged across measurements performed with different microendoscopes. Knowing the average number of pixels in between adjacent ticks in the image and the real distance between adjacent ticks in the sample (10 μm), we could determine, at each tick position, the local pixel size by dividing 10 μm by the mean number of pixels in between two adjacent ticks.

We observed that corrected microendoscopes had uneven magnification across the FOV (distor-tion), meaning that the local pixel size is not homogeneous all over the FOV. To characterize the uneven magnification independently of the acquisition setting (pixel size and microscope magnifi-cation), we computed the local magnification factor. This quantity was used to calibrate the local pixel size as a function of the radial distance. The local magnification factor was defined as the ratio

between the nominal pixel size and the local pixel size obtained as described above. The values of the local magnification factor as a function of the radial distance were fitted with a quartic function $f(x)=ax^4+bx^2+c$ for each microendoscope type (uncorrected and corrected) and the fitting function was used to calibrate the pixel size of all acquired images (*Supplementary file 3*). For each image acquired using corrected microendoscopes (e.g. *Figure 4*), we show the scale bar corresponding to the calibrated local pixel size at the center of the FOV. The radial change of pixel size in the *x* and *y* directions is visualized by a color-coded bar located close to the image and representing the local magnification factor normalized to the magnification factor at the center of the FOV. Additionally, considering the cumulative effect of the local change in the pixel size, we performed a calibration of radial distances longer than 10–20 µm, using the same ruler measurements described above. This calibration was obtained by fitting the nominal radial distances at which we identified the ruler ticks as a function of their real distances (given by the ruler spacing) with a quartic function $g(x)=dx^4+ex^2+fx+h$, for each microendoscope type (*Supplementary file 3*). The fitting function was then used to calculate the real distance among different structures across the FOV.

## Subresolved fluorescent beads

To quantify the extension of the PSF, we imaged subresolved fluorescent beads (FluoSpheres Carboxylate-Modified Microspheres, diameter: 100 nm, Cat. No. F8803, Thermo Fisher, Waltham, MA, US) monodispersed in a thin layer of poly-l-lysine (0.1 mg/mL, Cat. No. P2636, Merck, Darmstadt, Germany) on a microscope slide. Multiple z-stacks were acquired at 512 pixels x 512 pixels resolution (nominal pixel size: 0.049 µm/pixel) with 0.5 µm axial step and 8 frame averaging. Fluorescent beads were present across the FOV at different nominal radial distances from the optical axis (0–175 µm for corrected probes, 0–75 µm for uncorrected probes). In uncorrected microendoscopes, strong aberrations prevented imaging beads at nominal radial distances >75 µm. To quantify the axial resolution, for each bead we generated the *z*-axis intensity profile using the average value of the fluorescence signal within a ROI encompassing the whole bead perimeter after background subtraction and fitted this profile to either one or two Gaussian curves. For single Gaussian curves, we estimated the axial resolution as FWHM of the curve. In the case of fitting with two Gaussian curves, we computed the axial resolution as the sum of the half width at half maximum of the two Gaussian curves and the distance between their peaks. To quantify the lateral resolution, we generated the *x,y* intensity projections of individual beads and measured the intensity spatial distribution along eight profiles in eight different and equally spaced directions centered on the bead. We fitted each intensity profile to a Gaussian curve and we computed the average value of the FWHM of each curve as the lateral resolution. For each microendoscope type, measurements of axial and lateral resolution performed with multiple microendoscopes on multiple beads for each radial distance were pooled together to compute the mean value.

## Mouse fixed brain slices

To prepare brain slices, a deeply anesthetized mouse in which thalamic neurons were expressing jGCaMP7f was transcardially perfused first with PBS and then with 4% paraformaldehyde (PFA). The brain was harvested, fixed in 4% PFA overnight, and kept in sucrose (30 % v/v in PBS). Coronal sections (thickness: 50 µm) of the brain were cut using a microtome and kept in anti-freezing medium at –20 °C. On the day of the experiment, slices were washed three times with PBS for 10 min. One slice at a time was gently placed over a microscope slide and kept wet with drops of PBS along the duration of the experiment. On identified FOV containing neurons expressing the fluorescent indicator, we acquired a z-stack through the slice using a corrected or uncorrected microendoscope (1024 pixels x 1024 pixels resolution; nominal pixel size: 0.45 µm/pixel; axial step: 5 µm; number of axial steps: 23–32; frame averaging = 8).

## Generation of synthetic calcium imaging data

### Geometry of simulated neurons

For all simulated neural data (*Figures 5 and 6*, *Figure 6—figure supplements 1 and 2*), neurons were simulated as spheres with radius randomly sampled from a Gaussian distribution with mean and standard deviation (SD) estimated from the literature to be 10 µm and 3 µm, respectively. A nuclear region, of width randomly sampled from a normal distribution with mean = 5 µm and SD = 1 µm and

not expressing the fluorescent indicator, was added at the center of each simulated neuron. Therefore, only the spherical shell surrounding the nucleus generated the fluorescence signal. Simulated neurons were randomly placed with no overlap in a volume of size 400 × 400 × 170 μm³ and 360 × 360 × 200 μm³ for the 6.4 mm-long and the 8.8 mm-long microendoscopes, respectively, until neural density reached the value (26.8±7.2)·10⁴ cells/mm³ (**Suzuki and Bekkers, 2010**) or the volume was filled up with neurons. The resolution of the spatial volume was 0.8 μm/pixel in the *x* and *y* direction, 1 μm/pixel in the *z* direction.

## Simulated neural activity

Neural spiking activity was simulated for 5 min in 1ms time steps. Each neuron was randomly paired with other neurons (probability of two neurons of being paired equal to 0.05) to form groups of neurons with shared activity. The activity of each neuron was then simulated as the sum of an independent Poisson process and as many shared Poisson processes as the number of groups the neuron belonged to. The spike rate of the summed Poisson processes was the same for all neurons (0.3 Hz). These conditions resulted in average pairwise correlations of ground truth calcium activity larger than zero (average correlation 0.041±0.001, n=968130). This correlation value was roughly in the order of average pairwise correlations between neurons experimentally recorded with 2P calcium imaging (**Runyan et al., 2017**; **Valente et al., 2021**) and was independent of the radial distance. Calcium activity and fluorescence traces were finally generated using the equations and the parameters described for jGCaMP8f in **Zhang et al., 2023**.

## Generation of fluorescence t-series

The simulated FOV had dimensions 400 × 400 μm² and 360 × 360 μm² for the 6.4 mm-long and the 8.8 mm-long microendoscopes, respectively. The resolution was then adjusted according to the variations of the magnification factor, which were estimated from experimental data in **Figure 3E and F** and **Supplementary file 3**. This procedure led to a non-uniform resolution across the simulated FOV. To generate the synthetic calcium imaging t-series, we used the experimental measurements of spatial resolution reported in **Figure 3I and J** for both corrected and uncorrected microendoscopes. For corrected microendoscopes, the fifth-order polynomial function fitted to the experimental measurements of subresolved fluorescent films was rotated along the *z*-axis to generate the synthetic imaging focal surface. Moreover, the volume of fluorescence excitation was an ellipsoid resembling the experimentally measured PSF of corrected microendoscopes. For uncorrected microendoscopes, the synthetic imaging surfaces were two spherical shells with curvature radius estimated from the data (curvature radius: 273 μm and 2000 μm for the 6.4 mm-long uncorrected microendoscope and 992 μm and 2000 μm for the 8.8 mm-long uncorrected microendoscope) and the volume of fluorescence excitation was an ellipsoid resembling the experimentally measured PSF of uncorrected microendoscopes.

Simulated excitation volumes were synthetically scanned along the imaging focal surface (or surfaces), such that their axial direction was always orthogonal to the imaging focal surface(s). Due to the stronger field curvature of the 8.8 mm-long corrected microendoscope (**Figure 1C**) compared to 8.8 mm-long uncorrected microendoscopes, the center of the corrected imaging focal surface resulted at a larger depth in the simulated volume compared to the center of the uncorrected focal surface(s). Therefore, different simulated neurons were sampled in the two cases. We exactly followed methods previously published in **Antonini et al., 2020**. For completeness, the procedure description is repeated here. All voxels within the excitation volumes contributed to the signal of the corresponding pixel in the simulated FOV, as follows:

- If the pixel was at the FOV edge (radial distance >200 μm and >180 μm for the 6.4 mm-long and the 8.8 mm-long corrected microendoscopes, respectively), its signal was randomly sampled from a normal distribution, with mean and SD estimated from four experimental t-series recorded in vivo with the 8.8 mm-long corrected microendoscope and jGCaMP8f expressed in the mouse piriform cortex (see the paragraph '2P microscopy in the mouse olfactory cortex'). The temporal average intensities of pixels in the edges of the microendoscope FOVs, containing dark noise, were used to fit a Gaussian mixture model (component 1: proportion = 0.35; mean = 137.48; SD = 48.96; component 2: proportion = 0.65; mean = 126.83; SD = 5.02). The SD of the dark noise was assumed to depend on the dark noise mean in a linear way, with parameters

estimated from the same experimental data (intercept, $p_0$=–175.39; linear coefficient, $p_1$=1.57). The simulated dark noise was generated with the mean randomly sampled from the Gaussian Mixture Modeling (GMM) distribution and the SD proportional to the mean.

- If the pixel was in the central part of the FOV (radial distance ≤200 µm and ≤180 µm for the 6.4 mm-long and the 8.8 mm-long microendoscopes, respectively) but no neurons were within the excitation volume, the pixel signal was randomly sampled from a normal distribution with mean and SD estimated from experimental data. The mean intensity of pixels that were neither in the edges nor belonging to ROIs were fitted using a lognormal distribution (mean = 6.54, SD = 0.78) and the best linear fit between the squared root of the mean intensity and the intensity SD was computed ($p_0$=29.91, $p_1$=7.75). Simulated noise in the FOV was generated as Gaussian noise with mean randomly sampled from the lognormal distribution and SD proportional to the square root of the mean.

- If the pixel was in the central portion of the FOV (radial distance ≤200 µm and ≤180 µm for the 6.4 mm-long and the 8.8 mm-long microendoscopes, respectively) and at least one neuron was in the excitation volume(s), each voxel in the excitation volume(s) was assigned either Gaussian noise (estimated as in the previous condition) in case no neurons were in that voxel, or the fluorescence intensity of the neuron sampled by that voxel. If a neuron was contained within a voxel, Gaussian noise was also added to its signal. The mean of the added Gaussian noise was zero, while the SD was proportional to the square root of the mean intensity of the voxel, with the coefficients estimated from a linear fit between the square root of the mean intensity and the intensity SD of pixels assigned to ROIs in experimental data ($p_0$=322.18, $p_1$=0.98). The activity of all the voxels falling within the excitation volume(s) was then averaged to obtain the pixel's fluorescence intensity. The intensity of each pixel signal was finally modulated as a function of the radial position within the FOV, accordingly to the optical characterization of corrected and uncorrected microendoscopes using the radial intensity obtained by imaging the subresolved fluorescent films (*Figure 3*).

In simulations, the imaging rate of the t-series was set to 30 Hz.

## Analysis of simulated calcium t-series

### Detection of cell identities and extraction of activity traces

Analysis of synthetic calcium t-series was performed using CITE-ON (*Sità et al., 2022*) and custom code written in Python 3.7 (libraries: *NumPy*, https://numpy.org/doc/stable/; *SciPy*, https://docs.scipy.org/doc/scipy/, *Statsmodels*, https://www.statsmodels.org/stable/index.html). Specifically, cell bodies were detected and marked with rectangular bounding boxes, using the median intensity projection of the t-series as reference image. Fluorescence intensity traces of single neurons as a function of time, $F(t)$, were extracted using a custom modified version of the CITE-ON function *extract*. Within each rectangular bounding box and for each frame, this procedure selected as ROI the group of pixels with a fluorescence intensity value between the 80th and the 95th percentile of the fluorescence intensity distribution within the considered bounding box. $F(t)$ was computed frame-by-frame as the difference between the average signal of pixels in each ROI and the background signal. The background was calculated as the average signal of pixels that: (*i*) did not belong to any bounding box; (*ii*) had intensity values higher than the mean noise value measured in pixels located at the corners of the rectangular image, which do not belong to the circular FOV of the microendoscope; (*iii*) had intensity values lower than the maximum value of pixels within the boxes. Activity traces were computed as:

$$\frac{\Delta F}{F_o} = \frac{F(t) - F_0(t)}{F_0(t)},$$

where $F_0(t)$ is the cell baseline fluorescence at time $t$ estimated as the 20th percentile of the fluorescence distribution of the trace $F(t)$ in a temporal interval of 10 s centered in $t$. The peak SNR of activity traces $\Delta F/F_o$ was computed as:

$$\text{peak SNR} = \frac{max\left(\Delta F/F_o\right)}{noise},$$

where $max\left(\Delta F/F_o\right)$ is the maximum value of the activity trace and *noise* is the SD of the distribution of values below the 25th percentile of the intensity distribution of the entire trace (*Brondi et al., 2020*). To determine the position in the FOV of detected cells, we used the coordinates of the centroids of

the bounding boxes identified by CITE-ON (*Sità et al., 2022*), expressed in pixels. We converted the centroid coordinates in microns by multiplying the centroid coordinates by the nominal pixel size valid under undistorted conditions and then applied the distance magnification factor to correct for the FOV uneven magnification (see the subparagraph 'Calibration rulers'). In *Figure 5*, the number of detected ROIs and the maximum distance from the center of the FOV at which a cell identity was detected were computed as a function of the peak SNR threshold imposed on activity traces (tested peak SNR threshold values: 0, 5, 10, 11, 12.5, 15, 16.5, 17.5, 20, 22.5, 25, 27.5, 30).

## Pairwise correlation of adjacent neurons

In *Figure 6*, for each microendoscope type we computed the pairwise correlation of adjacent neurons as Pearson's correlation between the traces of any pair of nearby detected cells (distance between bounding box centroids ≤25 µm). Pearson's correlation was computed with the function *numpy.corrcoef*. To evaluate the contamination of pairwise correlation artefactually introduced by the spatial extension of the microendoscope excitation volume in uncorrected or corrected synthetic t-series, we estimated the expected cell pair correlation based on the simulated ground truth calcium activity as described below. We first computed the mean Pearson's correlation between the activity traces of any possible ground truth source pair for each simulated FOV. For each microendoscope type, we then defined the expected cell pair correlation as the mean Pearson's correlation of any possible ground truth source pair averaged over different FOV, plus three SD (see *Supplementary file 5* for parameters). In simulated t-series, any cell pair with pairwise correlation above the expected value could not be attributed to correlations between the ground truth calcium activity of source neurons and was expected to arise only because of the source mixing due to finite optical resolution of microendoscopic probes. Thus, we restricted the analysis to cell pairs with pairwise correlation higher than expected. We pooled together pairwise correlation data from FOV obtained for the same microendoscope type and we computed the percentage of detected adjacent cell pairs that had pairwise correlation higher than the expected cell pair correlation for different peak SNR thresholds. Pairwise correlation of adjacent cell pairs that exceeded the expected cell pair correlation threshold was also analyzed as a function of the radial distance of the cell pair, that was defined as the distance between the center of the FOV and the midpoint of the segment connecting the centroids of the two bounding boxes drawn by CITE-ON (*Sità et al., 2022*; *Figure 6—figure supplement 2A and C*).

## GLM of the contribution of ground truth source to activity traces

For each detected ROI of each simulated FOV, we considered the cellular ground truth sources overlapping in the imaging plane with the rectangular bounding box drawn by CITE-ON for at least 1 pixel. From now on, we define those sources as 'ground truth sources contributing to the extracted activity trace'. To evaluate the precision of microendoscopes in faithfully collecting the activity signals from individual cellular sources and avoiding source mixing, for each detected bounding box we applied a GLM of the ground truth sources contributing to its activity trace:

$$Y_i\left(t\right) = \sum_{j=1}^{k_i} a_j X_j\left(t\right) + q_i,$$

where: $Y_i\left(t\right)$ is the extracted activity trace for the i-th bounding box, with i=1,…, N, with N number of bounding boxes in the FOV with at least one contributing ground truth source; $k_i$ is the number of contributing ground truth sources for the i-th bounding box; $q_i$ is the i-th constant term; $X_j\left(t\right)$ is the activity trace of the j-th contributing ground truth source, with j=1,…, $k_i$; and $a_j$ is the j-th linear coefficient computed in the model. The GLM was applied using the methods *statsmodels.api.add_constant* to include the constant term and *statsmodels.genmod.generalized_linear_model.GLM.fit* (link function: *Identity*) to compute the $k_i$ linear coefficients. We evaluated the purity index of each extracted activity trace $Y_i\left(t\right)$ (i.e. the level of mixing of contributing sources) as:

$$\text{purity index} = \frac{\max_j\left(a_j^2\right)}{\sum_{j=1}^{k_i} a_j^2}$$

where $max_j()$ indicates the maximum value over j=1,…,$k_i$. We applied the GLM considering different peak SNR thresholds imposed on detected cell identities (*Figure 6—figure supplement 2B and D*).

## Correlation of extracted traces with ground truth source activity

For each detected bounding box with at least one contributing ground truth source and with peak SNR of the extracted trace >10, we computed the Pearson's correlation of the extracted trace with the activity trace of each contributing ground truth source. We sorted the sources in descending order of correlation and pooled together correlation values for the first (most correlated) ground truth source (*Figure 6D and I*). Correlation with the first ground truth source was analyzed as a function of the radial distance of the detected cell from the center of the FOV (*Figure 6E and J*).

## Animals

Experimental and surgical protocols were performed in accordance with the Guide of Care and Use of Laboratory Animals (NIH) and were approved by the Institutional Animal Care and Use Committee (IACUC protocol number: 24-03-0004) at Brown University and by the Istituto Italiano di Tecnologia Animal Health Regulatory Committee, by the National Council on Animal Care of the Italian Ministry of Health (authorization # 1134/2015-PR, # 689/2018-PR) and carried out according to the National legislation (D.Lgs. 26/2014) and to the legislation of the European Communities Council Directive (European Directive 2010/63/EU). C57BL/6 J mice were crossed to Ai14 mice (*Madisen et al., 2010*) and male and female heterozygous transgenic offspring of 8–12 weeks of age were used. Mice were maintained with unrestricted access to food and water under a 12 hr light/dark cycle and housed individually after surgery.

## Stereotaxic surgery for viral injections and microendoscope implantation

Viruses (pGP-AAV-syn-jGCaMP8f-WPRE, *Zhang et al., 2023*, or AAV1-syn-jGCaMP7f-WPRE, *Dana et al., 2019*) were purchased from Addgene (Watertown, MA, US) and injected using manually controlled pressure injection with a micropipette pulled and a micropipette puller (Sutter Instruments, Novato, CA, US). Mice were anesthetized with isofluorane with an induction at 3% and then maintained at 1–2% with an oxygen flow rate of ~1 L/min and head-fixed in a stereotactic frame (David Kopf, Tujunga, CA, US). Mice subcutaneously received buprenorphine slow release (0.05–0.1 mg/kg). Eyes were lubricated with an ophthalmic ointment and body temperature was stabilized using a heating pad attached to a temperature controller. Fur was shaved and the incision site was sterilized with isopropyl alcohol and betadine solution prior to beginning surgical procedures. A 1.0 mm round craniotomy was made using a dental drill centered to the following stereotaxic coordinates (mm): ML, 3.9; AP, 0.3. The virus solution diluted 1/3 in PBS was injected at speed 100 nL/min in three different spots (total injected volume: 1 µL) using the following coordinates (mm) to target olfactory cortex (ML/AP/DV): 3.85/0.6/–3.8, 3.95/0.3/–3.9, 4.05/0.0/–4.0, all relative to bregma (*Paxinos and Franklin, 2012*). After 5 min, the micropipette was slowly retracted from the brain at a speed of 500 µm/min. 30 min after viral injections, a corrected microendoscope was implanted above the olfactory cortex. The probe was implanted, centered to the craniotomy, at a speed of 100 µm/min until reaching the following coordinate: DV, –3.6. Once placed, the plastic ring of the endoscopic probe was fixed to the skull with Metabond adhesive cement (Parkell Inc, Edgewood, NY, US). A custom-made aluminum head bar was then attached to the skull using dental cement (Pi-ku-plast HP 36 Precision Pattern Resin, XPdent, Miami, FL, US). Finally, a protective cap made of Kwik-Sil silicone elastomer (World Precision Instruments, Sarasota, FL, US) was applied over the lens. Mice were allowed to recover from surgery for at least 6 weeks prior to imaging experiments. The criterion for inclusion of animal subjects in the study was based on the quality of the surgical implant, that is absence of blood and presence of >30 detectable cells in the maximally enlarged aberration corrected FOV of the microendoscope. Based on this criterion, six out of eight mice were used for subsequent experiments (average number of cells/FOV ± SD: 40±6, n=6 animals).

## Odor delivery

Animals were habituated to the experimenter and the head-fixation set-up for 30 min per day for at least two days before performing the imaging experiment. On imaging days, odor stimuli were

delivered through a custom built 16-channel olfactometer (Automate Scientific, Berkley, CA, US) equipped with a mass flow controller that maintained air flow at 1 L/min. The olfactometer solenoids were triggered by a Teensy 3.6 (PJRC). Vacuum was applied inside the 2P microscope isolation box to evacuate residual odor. For all experimental sessions, mice were habituated to the 2P microscope head-fixation set-up for 10 min prior to imaging. An odor trial lasted 22 s (5 s of prestimulus baseline, 2 ss of stimulation, 15 s of post-stimulus acquisition) with inter-trial intervals of 10 s. Odor stimuli were presented in pseudo-randomized fashion and 8 presentations of each odor were performed in a session. The odor panel consisted of (diluted in mineral oil v/v): acetophenone 1%, amylamine 1%, butyl acetate 1%, ethyl hexanoate 1%, 2-Isobutyl-3-methoxypyrazine 1%, β-Ionone 1%, 2,3-Pentanedione 0.1%, Valeric acid 0.1%. A photoionization detector (miniPID 200B, Aurora Scientific, Canada) was used to confirm reliable odor delivery.

## 2P microscopy in the mouse olfactory cortex

A typical imaging experiment lasted ~1.5 hr per mouse. 2P imaging of the olfactory cortex was performed using an Ultima Investigator laser scanning microscope (Bruker Nano, Inc, Middleton, WI, US) equipped with a 8 KHz resonance galvanometer and dual GaAsP PMTs (Cat. No. H10770, Hamamatsu, Hamamatsu City, Japan). Approximately 90–150 mW of laser power (at 920 nm, from Chameleon Discovery laser source [Coherent Inc, Santa Clara, CA, US]) was used during imaging, with adjustments in power levels to accommodate varying signals for each mouse. After focusing on the corrective lens surface using epifluorescence microscopy, optical viewing was switched to live view through the 2P laser, and a FOV was located by moving the objective ~100–200 µm upward. Images were acquired with a Nikon ×10 Plan Apochromat Lambda objective (0.45 NA, 4.0 mm WD) using a nominal pixel size of 0.72–0.84 µm/pixel. jGCaMP8f or jGCaMP7f signals were filtered through an ET-GFP (FITC/CY2) filter set. Acquisition speed was 30 Hz for 512 pixels × 512 pixels images.

## Analysis of in vivo 2P calcium imaging data

Analysis of 2P experimental t-series requiring detection of cell identities and trace extraction was performed using CITE-ON (*Sità et al., 2022*) and custom code written in Python 3.7, as for the synthetic t-series. Raw 2P t-series (duration: 8 minutes) were first motion-corrected using the CITE-ON function *motion_corr*, applied twice in succession when needed. Bounding boxes were detected using the median intensity projection of the motion-corrected t-series as reference. Activity traces, peak SNR and Pearson's correlations were computed as described for synthetic calcium data. Pairwise correlation of adjacent neurons was explored considering nearby detected cells for which the distance between the bounding boxes centroids was either ≤25 µm or ≤30 µm (*Supplementary file 6*).

## Statistics

Statistical analysis was performed using Matlab for optical characterization data and using the Python library of statistical functions *scipy.stats* (https://docs.scipy.org/doc/scipy/reference/stats.html) for synthetic and experimental calcium imaging data. The number n of repeated measurements and the number m of experimental replicates are indicated in figure legends. Normality of data distributions was tested using either Shapiro-Wilk test for small sample sizes ($n<50$) or D'Agostino-Pearson test ($n\geq50$). The threshold for statistical significance was set at $p=0.05$ and indicated as follows: *, $p<0.05$, **, $p<0.01$, ***, $p<0.001$. p values higher than 0.05 were considered not significant (n.s.). Data are presented as mean values ± standard error of the mean (s.e.m.), unless otherwise stated. Linear regressions were performed using the function *scipy.stats.linregress*. To test the significance of linear fit slope being different from zero, we used the Wald test with *t*-distribution in case the fit residuals were normally distributed and the permutation test with the function *stats.permutation_test* if the fit residuals were not normally distributed. The null distribution of linear coefficients used in the permutation test was built performing multiple linear regressions on a subset of possible permutations of data (permutation type: 'pairings', $10^4$ permutations for $n<500$, $10^5$ permutations for $n\geq500$). To test the significance of linear fit slopes being different from each other, we used a permutation test in which the observations were randomly assigned to the two samples (corrected or uncorrected). The null distribution was built calculating the difference between the two slopes for a subset of possible permutation of data ($10^4$ permutations for $n<500$, $10^5$ permutations for $n\geq500$). To test the significance of the difference between

two means of non-normally distributed samples, we used a permutation test. In this case, the null distribution was built calculating the difference between the two mean values for a subset of possible permutations of data (permutation type: 'independent', $10^4$ permutations for n<500, $10^5$ permutations for n≥500).

## Acknowledgements

We thank Celeste Bortolani, Doriana Debellis, Fabio Moia, and Francesca Succol for technical support and members of the Fellin and Panzeri laboratories for useful discussions and suggestions. This work was supported in part by Horizon 2020 ICT (https://cordis.europa.eu/project/id/101016787, DEEPER) and Next Generation EU (https://www.fondazione-fair.it/, FAIR). Work in the Fleischmann lab was funded by the National Institutes of Health (NIDCD 1R01DC017437).

## Additional information

### Funding

| Funder | Grant reference number | Author |
| --- | --- | --- |
| Horizon 2020 Framework Programme | 10.3030/101016787 | Tommaso Fellin |
| European Commission | NextGenerationEU FAIR PE0000013 | Tommaso Fellin |
| National Institute on Deafness and Other Communication Disorders | 1R01DC017437 | Alexander Fleischmann |

The funders had no role in study design, data collection and interpretation, or the decision to submit the work for publication.

### Author contributions

Andrea Sattin, Software, Formal analysis, Validation, Investigation, Visualization, Methodology, Writing – review and editing; Chiara Nardin, Data curation, Software, Formal analysis, Validation, Investigation, Visualization, Writing – original draft, Writing – review and editing; Simon Daste, Visualization, Methodology, Writing – original draft, Writing – review and editing; Monica Moroni, Formal analysis, Methodology, Writing – original draft; Innem Reddy, Methodology; Carlo Liberale, Alexander Fleischmann, Resources, Supervision, Writing – original draft, Writing – review and editing; Stefano Panzeri, Conceptualization, Resources, Supervision, Validation, Methodology, Writing – original draft, Writing – review and editing; Tommaso Fellin, Conceptualization, Resources, Supervision, Funding acquisition, Validation, Investigation, Visualization, Methodology, Writing – original draft, Project administration, Writing – review and editing

### Author ORCIDs

Andrea Sattin ⓘ https://orcid.org/0000-0002-1345-2204
Chiara Nardin ⓘ https://orcid.org/0000-0002-4084-4932
Carlo Liberale ⓘ https://orcid.org/0000-0002-5653-199X
Stefano Panzeri ⓘ https://orcid.org/0000-0003-1700-8909
Alexander Fleischmann ⓘ https://orcid.org/0000-0001-7956-9096
Tommaso Fellin ⓘ https://orcid.org/0000-0003-2718-7533

### Ethics

Experimental and surgical protocols were performed in accordance with the Guide of Care and Use of Laboratory Animals (NIH) and were approved by the Institutional Animal Care and Use Committee (IACUC protocol number: 24-03-0004) at Brown University and by the Istituto Italiano di Tecnologia Animal Health Regulatory Committee, by the National Council on Animal Care of the Italian Ministry of Health (authorization # 1134/2015-PR, # 689/2018-PR) and carried out according to the National legislation (D.Lgs. 26/2014) and to the legislation of the European Communities Council Directive (European Directive 2010/63/EU).

Reviewer #1 (Public review): https://doi.org/10.7554/eLife.101420.4.sa1
Reviewer #2 (Public review): https://doi.org/10.7554/eLife.101420.4.sa2
Reviewer #3 (Public review): https://doi.org/10.7554/eLife.101420.4.sa3
Author response https://doi.org/10.7554/eLife.101420.4.sa4

## Additional files

### Supplementary files

Supplementary file 1. Parameters of the polynomial function describing the aspherical surface of simulated corrective lenses. Parameters of *Equation 1* (see Materials and Methods) for the simulated corrective lens designed to be applied to the GRIN rods of length 6.4 mm (top row) and 8.8 mm (bottom row).

Supplementary file 2. Spatial resolution of simulated uncorrected and corrected microendoscopes. Axial and lateral resolution of simulated microendoscopes were evaluated measuring the dimensions of simulated 2P PSF for each probe at different radial distances. $x,z$ (Axial) and $x,y$ (Lateral) intensity profiles of simulated PSFs were fitted with Gaussian curves and their FWHM was used to define the resolution, as done for experimental PSFs (see Materials and Methods).

Supplementary file 3. Parameters used for the computation of the local pixel size and for distance calibration of images acquired with microendoscopes. Coefficients of the quartic functions fitting the measurements performed on images acquired on the calibration ruler for uncorrected and corrected microendoscopes based on the 6.4 mm-long GRIN rod (left) and the 8.8 mm-long GRIN rod (right). The numbers in parenthesis indicate the 95% lower and upper confidence bounds (see *Figure 3E and F*). R-square values are indicated for each fit.

Supplementary file 4. Fitting parameters for PSF measurements of uncorrected and corrected microendoscopes. Coefficients of quartic functions fitting experimental PSF data (axial, top; lateral, bottom) are presented for uncorrected and corrected microendoscopes based on the 6.4 mm-long GRIN rod (left) and the 8.8 mm-long GRIN rod length (right). Parentheses indicate the 95% lower and upper confidence bounds (see *Figure 3I and J*). R-square values are indicated for each fit.

Supplementary file 5. Expected Pearson's correlation of cell pair in synthetic calcium data. Numerical values used to estimate the expected correlation between cell pairs in synthetic calcium t-series are indicated for each microendoscope type. The table displays the mean and SD of Pearson's correlation between the activity traces of any possible ground truth source neuron pair obtained from $n$ simulated FOVs and the expected cell pair correlation (mean Pearson's correlation plus three SDs). These parameters were used for the analysis in *Figure 6A and F* and in *Figure 6—figure supplement 2A and C*.

Supplementary file 6. Linear regression analysis for pairwise correlation of adjacent neurons as a function of the radial distance of pair centroid for in vivo 2P imaging data. The values of the slope of the linear fits are indicated ± s.e. for adjacent neurons with maximum centroid distance equal to 25 μm (top) or 30 μm (bottom). Results obtained with jGCaMP8f, jGCaMP7f, and with the merged dataset including both jGCaMP8f and jGCaMP7f are displayed. The number $n$ of adjacent neuron pairs, the $p$ value of the indicated statistical test for the normality of residuals, and the $p$ value of the Wald test on the null hypothesis of slope = 0 are indicated for each condition.

MDAR checklist

### Data availability

The softwares SyntheticTseriesGeneration used in this paper to generate artificial t-series (Figure 5, 6 and Figure 6-figure supplement 1, 2), ExperimentalAndSyntheticTseries used to analyze artificial and experimental t-series (Figure 7 and 8), and OpticalCharacterization used to analyze optical measurements of subresolved fluorescent films and beads (Figure 3) are available on GitHub (copy archived at *Sattin et al., 2024*). Datasets containing raw synthetic calcium data related to Figures 5, 6 and Figure 6-figure supplements 1, 2 are available in Zenodo public repositories at the following links:https://zenodo.org/records/15206281 for the 6.4 mm-long corrected microendoscope; https://zenodo.org/records/15210027 for the 6.4 mm-long uncorrected microendoscope;https://doi.org/10.5281/zenodo.15212139 for the 8.8 mm-long corrected microendoscope;https://doi.org/10.5281/zenodo.15212733 for the 8.8 mm-long uncorrected microendoscope.

The following datasets were generated:

| Author(s) | Year | Dataset title | Dataset URL | Database and Identifier |
|-----------|------|---------------|-------------|-------------------------|
| Moroni M, Nardin C, Sattin A, Panzeri S, Fellin T | 2025 | Synthetic calcium data for the 6.4 mm-long corrected microendoscope | https://zenodo.org/records/15206281 | Zenodo, 10.5281/zenodo.15206281 |
| Moroni M, Nardin C, Sattin A, Panzeri S, Fellin T | 2025 | Synthetic calcium data for the 6.4 mm-long uncorrected microendoscope | https://zenodo.org/records/15210027 | Zenodo, 10.5281/zenodo.15210027 |
| Moroni M, Nardin C, Sattin A, Panzeri S, Fellin T | 2025 | Synthetic calcium data for the 8.8 mm-long corrected microendoscope | https://zenodo.org/records/15212139 | Zenodo, 10.5281/zenodo.15212139 |
| Moroni M, Nardin C, Sattin A, Panzeri S, Fellin T | 2025 | Synthetic calcium data for the 8.8 mm-long uncorrected microendoscope | https://zenodo.org/records/15212733 | Zenodo, 10.5281/zenodo.15212733 |

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
