## [Editor Report · eLife Assessment]

This **valuable** study builds on previous work by the authors by presenting a potentially key method for correcting optical aberrations in GRIN lens-based microendoscopes used for imaging deep brain regions. By combining simulations and experiments, the authors provide **convincing** evidence showing that the obtained field of view is significantly increased with corrected, versus uncorrected microendoscopes. Because the approach described in this paper does not require any microscope or software modifications, it can be readily adopted by neuroscientists who wish to image neuronal activity deep in the brain.

---

## [Referee Report · Reviewer #1 (Public review)]

Summary:

Sattin, Nardin, and colleagues designed and evaluated corrective microlenses that increase the useable field of view of two long (>6mm) thin (500 um diameter) GRIN lenses used in deep-tissue two-photon imaging. This paper closely follows the thread of earlier work from the same group (esp. Antonini et al, 2020; eLife), filling out the quiver of available extended-field-of-view 2P endoscopes with these longer lenses. The lenses are made by a molding process that appears practical and easy to adopt with conventional two-photon microscopes.

Simulations are used to motivate the benefits of extended field of view, demonstrating that more cells can be recorded, with less mixing of signals in extracted traces, when recorded with higher optical resolution. In vivo tests were performed in piriform cortex, which is difficult to access, especially in chronic preparations.

The design, characterization, and simulations are clear and thorough, but they do not break new ground in optical design or biological application. However, the approach shows much promise, including for applications such as miniaturized GRIN-based microscopes. Readers will largely be interested in this work for practical reasons: to apply the authors' corrected endoscopes to their own research.

Strengths:

The text is clearly written, the ex vivo analysis is thorough and well supported, and the figures are clear. The authors achieved their aims, as evidenced by the images presented, and were able to make measurements from large numbers of cells simultaneously in vivo in a difficult preparation.

The authors did a good job of addressing issues I raised in initial review, including analyses of chromaticity and the axial field of view, descriptions of manufacturing and assembly yield, explanations in the text of differences between ex vivo and in vivo imaging conditions, and basic analysis of the in vivo recordings relative to odor presentations. They have also shortened the text, reduced repetition, and better motivated their approach in the introduction.

---

## [Referee Report · Reviewer #2 (Public review)]

In this manuscript, the authors present an approach to correct GRIN lens aberrations, which primarily cause a decrease in signal-to-noise ratio (SNR), particularly in the lateral regions of the field-of-view (FOV), thereby limiting the usable FOV. The authors propose to mitigate these aberrations by designing and fabricating aspherical corrective lenses using ray trace simulations and two-photon lithography, respectively; the corrective lenses are then mounted on the back aperture of the GRIN lens.

This approach was previously demonstrated by the same lab for GRIN lenses shorter than 4.1 mm (Antonini et al., eLife, 2020). In the current work, the authors extend their method to a new class of GRIN lenses with lengths exceeding 6 mm, enabling access to deeper brain regions as most ventral region of the mouse brain. Specifically, they designed and characterized corrective lenses for GRIN lenses measuring 6.4 mm and 8.8 mm in length. Finally, they applied these corrected long micro-endoscopes to perform high-precision calcium signal recordings in the olfactory cortex.

Compared with alternative approaches using adaptive optics, the main strength of this method is that it does not require hardware or software modifications, nor does it limit the system's temporal resolution. The manuscript is well-written, the data are clearly presented, and the experiments convincingly demonstrate the advantages of the corrective lenses.

The implementation of these long corrected micro-endoscopes, demonstrated here for deep imaging in the mouse olfactory bulb, will also enable deep imaging in larger mammals such as rats or marmosets.

Comments on revisions:

The authors have clearly addressed all my comments.

---

## [Referee Report · Reviewer #3 (Public review)]

Summary:

This work presents the development, characterization and use of new thin microendoscopes (500µm diameter) whose accessible field of view has been extended by the addition of a corrective optical element glued to the entrance face. Two microendoscopes of different lengths (6.4mm and 8.8mm) have been developed, allowing imaging of neuronal activity in brain regions >4mm deep. An alternative solution to increase the field of view could be to add an adaptive optics loop to the microscope to correct the aberrations of the GRIN lens. The solution presented in this paper does not require any modification of the optical microscope and can therefore be easily accessible to any neuroscience laboratory performing optical imaging of neuronal activity.

Strengths:

(1) The paper is generally clear and well written. The scientific approach is well structured, and numerous experiments and simulations are presented to evaluate the performance of corrected microendoscopes. In particular, we can highlight several consistent and convincing pieces of evidence for the improved performance of corrected microendoscopes:

- PSFs measured with corrected microendoscopes 75µm from the centre of the FOV show a significant reduction in optical aberrations compared to PSFs measured with uncorrected microendoscopes.

- Morphological imaging of fixed brain slices shows that optical resolution is maintained over a larger field of view with corrected microendoscopes compared to uncorrected ones, allowing neuronal processes to be revealed even close to the edge of the FOV.

- Using synthetic calcium data, the authors showed that the signals obtained with the corrected microendoscopes have a significantly stronger correlation with the ground truth signals than those obtained with uncorrected microendoscopes.

(2) There is a strong need for high quality microendoscopes to image deep brain regions in vivo. The solution proposed by the authors is simple, efficient and potentially easy to disseminate within the neuroscience community.

Weaknesses:

Weaknesses that were present in the first version of the paper were carefully addressed by the authors.

---

## [Author Response]

The following is the authors’ response to the previous reviews

**Reviewer #1:**
(1) As discussed in review and nicely simulated by the authors, the large figure error indicated by profilometry (~10 um in some cases on average) is inconsistent with the optical performance improvements observed, suggesting that those measurements are inaccurate.I see no reason to include these inaccurate measurements.

We agree with the Referee and removed the indicated figure (old Supplementary Fig. 4) and data.

**Reviewer #3:**
(1) It would be interesting to comment on how the addition of a coverslip changes the performance of the uncorrected microendoscope compared to the use of bare grin lenses.

We modified the discussion section (page 18) and added a new reference (#36) to include the request of the Referee.

(2) In Figure 6C-H, the authors can indeed show data corresponding to all detected cells, but I still think that the statistics should be calculated using the same effective FOV.

We modified Figure 6 legend to include the request of the Referee.

(3) Authors could present the images in Figures 4-6 as in the original version, with a scale bar in the centre of the FOV that is different for the two types of objectives (corrected vs uncorrected). They could add a short justification for this choice, and perhaps present the other version for Figure 4 in a supplementary information sheet (with similar scale bars at the centre of the FOV for both types of objectives). It would allow readers to appreciate that the FOV still appears significantly enlarged with this other presentation.

As requested by the Referee, we modified the text in the Result section (page 11) and added the additional version of Figure 4 as Figure 4-figure supplement 1.